# Super-Enhancers and Their Parts: From Prediction Efforts to Pathognomonic Status

**DOI:** 10.3390/ijms25063103

**Published:** 2024-03-07

**Authors:** Anastasia V. Vasileva, Marina G. Gladkova, German A. Ashniev, Ekaterina D. Osintseva, Alexey V. Orlov, Ekaterina V. Kravchuk, Anna V. Boldyreva, Alexander G. Burenin, Petr I. Nikitin, Natalia N. Orlova

**Affiliations:** 1Prokhorov General Physics Institute of the Russian Academy of Sciences, 38 Vavilov St., 119991 Moscow, Russia; anastasia.vasileva.work@gmail.com (A.V.V.); marinegladkova@gmail.com (M.G.G.); escobar.morente@gmail.com (G.A.A.); osintseva.ed@gmail.com (E.D.O.); kravchuk.ev.15@gmail.com (E.V.K.); anna.v.pushkareva@gmail.com (A.V.B.); agburenin@gmail.com (A.G.B.); nikitin@kapella.gpi.ru (P.I.N.); 2Faculty of Bioengineering and Bioinformatics, Lomonosov Moscow State University, GSP-1, Leninskiye Gory, MSU, 1-73, 119234 Moscow, Russia; 3Faculty of Biology, Lomonosov Moscow State University, Leninskiye Gory, MSU, 1-12, 119991 Moscow, Russia

**Keywords:** enhancers, super-enhancers, dynamics, locus-control regions, chromatin structure regulation, chromatin interactions

## Abstract

Super-enhancers (SEs) are regions of the genome that play a crucial regulatory role in gene expression by promoting large-scale transcriptional responses in various cell types and tissues. Recent research suggests that alterations in super-enhancer activity can contribute to the development and progression of various disorders. The aim of this research is to explore the multifaceted roles of super-enhancers in gene regulation and their significant implications for understanding and treating complex diseases. Here, we study and summarise the classification of super-enhancer constituents, their possible modes of interaction, and cross-regulation, including super-enhancer RNAs (seRNAs). We try to investigate the opportunity of SE dynamics prediction based on the hierarchy of enhancer single elements (enhancers) and their aggregated action. To further our understanding, we conducted an in silico experiment to compare and differentiate between super-enhancers and locus-control regions (LCRs), shedding light on the enigmatic relationship between LCRs and SEs within the human genome. Particular attention is paid to the classification of specific mechanisms and their diversity, exemplified by various oncological, cardiovascular, and immunological diseases, as well as an overview of several anti-SE therapies. Overall, the work presents a comprehensive analysis of super-enhancers across different diseases, aiming to provide insights into their regulatory roles and may act as a rationale for future clinical interventions targeting these regulatory elements.

## 1. Introduction

All human somatic cells contain the same DNA nucleotide sequence. The existence of about 200 different cell lines in the human body [1] is mainly explained by the possibility of differential expression patterns of 19,969 functional genes [2]. As a result, the proteins necessary for cellular identity are synthesised [3]. To organise the normal development and functioning of a cell, strict qualitative and quantitative control of gene transcription is required. This is carried out by certain regulatory sections of DNA known as cis-regulatory elements, which include promoters, enhancers, silencers, and insulators [4]. Enhancers in particular demonstrate the greatest variability between cell types [1], suggesting their important role in controlling tissue-specific and developmental genes [3,5].

Active enhancers are open chromatin regions enriched with transcription factors (TFs), cofactors, and active chromatin marks [6]. The main function of enhancers is to increase the expression of regulated genes. The regulation of cell-identity genes has been associated with enhancer clusters called super-enhancers (SEs). SEs are characterised by their length and higher concentration of master TFs, mediator, p300, cohesin, H3K27ac, and H3K4me1 marks compared to typical enhancers [7].

SEs have been classified as a separate class of cis-regulatory elements along with locus-control regions and stretch enhancers. However, this classification remains disputable. Studies have investigated SEs based on their association with developmentally important genes, structural features, interactions between SE subunits, and their connections to diseases [8,9,10,11,12].

While several studies have investigated SEs based on their association with developmentally important genes, structural features, and disease connections, there remains a need to review and consolidate the accumulated results of SE research. This work aims to provide a comprehensive summary of the research on SEs over the past decade since their discovery. By examining the distinctive features of SEs, including their unique traits compared to locus-control regions (LCRs) and stretch enhancers, classification attempts, formation mechanisms, functioning models, and identification methods, we aim to fill the existing knowledge gaps and provide a comprehensive understanding of SEs. Additionally, we will discuss the numerous studies available to date that link SE dynamics to oncological, inflammatory, and other diseases. By exploring the implications of these findings, we will contemplate several potential anti-SE therapies. Through this review, we aim to contribute to the current understanding of SEs and their significance in gene regulation and disease pathogenesis.

## 2. Super-Enhancers: Historical Background, Characteristics, and Search Methods

### 2.1. Enhancer Clusters—SE Concept Development History

In 1985, while studying the upstream sequence of the human β-like-globin gene cluster, Tuan et al. found that ε-globin regulation in erythroid cells is carried out by several enhancers at once [13]. Subsequently, it was shown that similar enhancer clusters are involved in the regulation of a number of other tissue-specific genes in mammals, increasing their expression to physiological levels in a copy-dependent manner [14]. Such clusters were named locus-control regions (LCRs).

In 2013, researchers from Young’s laboratory [7] studied the causes of expression dysregulation of cell-identity genes after a decrease in the concentration of the mediator complex (MC), as previously observed by Kagey et al. [15]. The researchers found that in mESCs, 40% of MCs associated with enhancers in pro-B cells were bound to some large enhancer clusters, the length of which was an order of magnitude greater than the length of typical enhancers (TEs) [7]. The exploration of these enhancer clusters led the authors to identify their core characteristics. Namely, mentioned enhancer clusters:Regulate cell-identity genes, including master transcription-factor (TF) genes (it is worth noting that SE clusters were found to not be associated with housekeeping genes);Have an order of magnitude higher concentration of TF, MC, and chromatin modification marks (H3K27ac and H3K4me1) compared to TEs;Have an order of magnitude greater length compared to TEs;Are sensitive to changes in the concentrations of MC and master TF;An enhancer cluster contains only those enhancers that are located no further than 12.5 kb from each other (a definition assumption made after mESCs analysis) [7].

Based on these observations, the authors speculated that a new regulatory element was found and assigned it a separate name, “super-enhancers” (SEs). In this review, we will adhere to the definition of SE provided by the researchers in Young’s laboratory.

Expanding on the study of White et al., several authors who examined the molecular environment of SEs, in addition to master TF and MC, also found high concentrations of bromodomain and extraterminal domain (BET) proteins (such as BRD4) [16], RNA polII, cohesin, Nipbl, p300, CBP, Chd7, Brd4, and components of esBAF and Lsd1-Nurd complexes [17]. Also, studies have shown that super-enhancers can influence miRNA expression and function. Integrative microRNA network analysis has revealed the connectivity between super-enhancers and miRNAs, providing insights into tissue-specific miRNA expression, miRNA function, and evolutionary aspects of miRNA regulation. The specific mechanisms by which SEs influence miRNA processing are still being investigated. However, it is clear that SEs contribute to the regulation of miRNA expression and function, adding another layer of complexity to the understanding of miRNA biology [18].

The existence of SEs has been confirmed in numerous cell lines and tissues, and it has been shown to be often cell-type specific. Specifically, SEs were found in mouse pro-B cells (master TF PU.1), myotubes (MyoD), T helpers (Th T-Bet), macrophages (C/EBPa) [7], 86 human cell lines and tissues [17], other mammalian vertebrates (such as pigs), nonmammalian vertebrates (zebrafish) [19], invertebrates (*Caenorhabditis elegans*) [20] and plants (*Arabidopsis thaliana*) [21,22], and unicellular eukaryotes (*Saccharomyces cerevisiae*) [23]. This supports the idea that SEs play a special role in cell development and lineage commitment.

Almost simultaneously, Parker et al., while studying enhancers in 10 cell lines, discovered continuous enhancers (≥3 kb in length) located in the same neighbourhood as genes responsible for cell identity and enriched for disease-associated mutations [24]. Suggesting that these enhancers have a special function, the authors introduced the term “stretched enhancer” to designate them.

To the best of our knowledge, no systematic comparison between LCRs, SEs, and stretched enhancers has been published. However, Khan et al. performed part of this comparative analysis, focusing on the differences and similarities between SEs and stretched enhancers considering cell-type specificity [25]. The intersection of those regulatory elements showed that 85% of SEs overlapped with only 13% of stretched enhancers. Considering a number of features associated with enhancer activity (Table 1), it was concluded that most stretched enhancers are poised and less active in comparison to SEs. The summarised results of the SE-stretched enhancer comparison are presented in Table 1.

In addition to that, some of the known LCRs overlap SEs (including LCRs associated with beta-globin locus (K562), OCT4 (hESC), IFNAR1 (foetal thymus)) [17], and stretched enhancers [24]. Super-enhancers (SEs) were chosen to be emphasised in the current work because they are the subject of active and ongoing research, and the term “super-enhancer” became more popular in academic circles, compared to “LCR” and “stretch enhancers”. However, some hybrid terms have been noted in the literature, e.g., “super stretchy enhancers” [26].

Further, we will address the following key questions: existing hypotheses concerning the structure of SEs, including interactions among SE constituents, the 3D structure and hierarchy of SEs, mechanisms of SE–promoter interactions, the function of seRNA, and the correlation between SEs and various diseases, including cancer and inflammatory diseases.

### 2.2. SE–Promoter Interaction and SE-Mediated Gene-Activation Models

The formation of SEs is a highly sophisticated and meticulously orchestrated process within the realm of gene regulation. SEs emerge as a result of the cooperative engagement of an array of transcription factors and cofactors, taking place in the dynamic context of chromatin. Within the intricate chromatin landscape, multiple enhancer elements densely populate vast genomic regions, establishing complex interactions with an array of regulatory proteins. These expansive enhancer clusters, which represent nascent SEs, are characterised by their remarkable length and their ability to recruit elevated concentrations of transcriptional machinery (the distinguishing hallmarks of SEs). It is believed that their formation is not merely the product of individual actions by transcription factors but, rather, a collective endeavour to create high-density enhancer landscapes. It is important to note that a comprehensive review of the mechanisms governing SE assembly in oncologies has been proposed in [27]. While the mechanisms of SE formation are indeed complex and multifaceted, they are only part of the larger narrative.

The true essence of SE functionality lies in SEs’ interactions with target gene promoters, a process instrumental in shaping gene expression in normal cells and often being disrupted in cells under stress. SEs’ target operation is thought to be dependent on the strength of topologically associated domain (TAD) boundaries [28]. Indeed, it has been demonstrated that TADs containing SEs typically have stronger boundaries [29]. In this context, the strength of a boundary is to be understood as the frequency of inter-TAD interactions [30], where a low frequency of interactions implies robust TAD boundaries, which correlates with the concentration of structural proteins such as CCCTC-binding factors (CTCF) at their boundaries [31]. Additionally, TADs contain substructures known as sub-TADs. They confine the given super-enhancer’s sphere of influence to its target gene(s). Structurally, sub-TADs include the SE along with the target gene(s) and are demarcated by CTCF insulation borders, co-occupied by cohesin on both sides (Figure 1). The median size of sub-TADs is typically around 185–200 kb [32], whereas super-enhancer domains encompass an average of 1–2 genes, and SE domains have an average length takes up to 106 kb [33].

Sub-TADs are organised into small loops, formed through the interaction of CTCF–CTCF borders [33]. The functional relevance of sub-TADs was experimentally confirmed by Dowen et al. The researchers deleted one of the sub-TAD borders, which contained the SE associated with genes critical for ESCs’ identity, such as miR-290-295 gene group, *NANOG*, *TDGF1*, *POU5F1* (Oct4), and *PRDM14*. As expected, this deletion led to changes in the expression of target genes as well as genes located at the external borders of the removed CTCF sites. The authors discovered that the majority of SEs (82%) do not span sub-TAD borders.

However, while sub-TADs are crucial for ensuring high specificity in SE–promoter (SE-P) interactions, they are not the sole determinants. It is assumed that additional specificity might be provided by the unique landscape of TFs and cofactors at the target promoter(s) that are recognised by the SE [34]. Indeed, in the experiments conducted by Zeng et al., the deletion of the insulated border separating *Wap*-SE from its target gene did not lead to the activation of the latter. Furthermore, the fusion of *Wap*-SE with the nontarget gene Tbrg4 resulted in a fourfold increase in Tbrg4 expression, whereas when the fusion of *Wap*-SE together with *TBRG4* promoter occurred, Tbrg4 expression was elevated by 80 times. As mentioned earlier, these findings allow us to speculate on the importance of the promoter protein landscape in SE–promoter interactions [34].

To add to the existing complexity of SE-P recognition specificity, leaky sub-TADs were discovered, which allows us to speculate that the strength of sub-TADs’ borders is an additional regulatory layer during the processes of development, lineage commitment, and response to external stimulus. Although, in most of the cases, SE activity is, in fact, limited by sub-TADs, this statement was shown to have exceptions. In the research performed by Vos et al., it was found that in ESCs one of the SEs (*Prdm14*-SE) simultaneously regulates two genes belonging to different sub-TADs (*SLCO5A1* and *PRM14*). Interestingly, under normal conditions, it increases Slo5a1 expression by 2 times, compared to 12 times, when the insulation sites are deleted. Therefore, this demonstrates that the strength of insulation can influence gene expression [35].

In most cases, SE regulates the expression of only one gene [36]. However, there are examples of SEs regulating a gene cluster [37,38,39]. In such instances, the question arises about the mechanism behind the activation of multiple promoters by a single SE.

To address this question, two models have been proposed: the competitive (flip-flop) model and the cooperative model (Figure 2). According to the competitive model, promoters of genes regulated by one SE compete for binding to the SE [40,41], while in the cooperative model, their activation occurs simultaneously [39]. Both models have received experimental confirmation.

Competitive promoter activation has been observed in fast *Myosin* genes in adult skeletal muscle lineage, where it controls myofiber identity [37], as well as in the *Nppa* and *Nppb* genes [38]. Conversely, cooperative activation of promoters has been noted in α-globin genes during erythroid differentiation [39] and β-globin genes [42]. Moreover, in B-cells, *IgH locus* promoter genes regulated by the same SE (3’RR) can be activated in both ways (competitively and cooperatively) depending on the inducing signal. This process is crucial for immunoglobulin isotype switching. It is worth noting that, during the cooperative activation of promoters, the activity of SEs can be unevenly distributed among promoters, which may be attributed to the relative positioning of super-enhancers and promoters [43].

### 2.3. Hierarchy of SE Elements

SEs are thought to be composed of constituent enhancers, which are smaller regulatory elements that work together to form the SE. These constituent enhancers are stitched together within the SE, with gaps of up to 12.5 kb between them [9]. The identification and grouping of these constituent enhancers to assign a super-enhancer to a target gene are done using bioinformatics algorithms, such as the ROSE software (https://github.com/stjude/ROSE, accessed on 25 February 2024) [44].

The constituents of SEs might also be regarded as concomitant to single enhancer elements transcription factors, cofactors, chromatin regulators, and the transcription apparatus. These factors occupy the SEs and contribute to their regulatory function [17]. SEs are densely occupied by the transcription apparatus and its cofactors, including cohesins, which play a role in regulating gene expression through gene loops and CTCF-mediated interactions [45].

The interaction and cross-regulation of super-enhancer constituents are facilitated by their capacity to drive short- and long-range interactions through phase separation and 3D genomic association [9]. This unique property of SEs distinguishes them from typical enhancers and contributes to their ability to activate transcription to a greater extent than the sum of their constituent enhancers [9].

Almost immediately after the discovery of SEs, scientists endeavoured to identify the roles of individual elements within SEs and their contributions to overall SE activity. It has been demonstrated that many SEs exhibit functional and temporal hierarchies.

In a study conducted by Huang et al., utilising Hi-C and ChIP-seq methods, the authors revealed that approximately one-third of SEs exhibit hierarchical organisation, comprising hub and non-hub enhancers (Figure 3A).

Hub enhancers share similar active markers with non-hub enhancers; however, they are notably enriched in binding sites for structural proteins, specifically CTCF and cohesin. It has been conjectured that hub enhancers may play a critical role in the structural organisation of SEs. Indeed, the removal of a hub enhancer has been observed to locally disrupt chromatin organisation and reduce the expression of the regulated gene. Furthermore, it is worth mentioning that mutations within hub enhancers have been associated with various diseases [46].

Another type of hierarchy among SE elements was demonstrated by Hay et al. [47]. While studying α-globin SE in mouse erythroid cells, the authors observed variations in the concentration of TFs and MCs across different SE elements. Specifically, only two enhancers, R1 and R2, exhibited enrichment in MCs and were associated with all four master TFs considered in the study (Nf-e2, Gata1, Scl, and Klf1). In contrast, other enhancers displayed lower concentrations of MC and were associated with only two to three of the master TFs. Accordingly, these enhancers were classified as “strong” and “weak” (Figure 3B).

Experimental dissection of the SE demonstrated that strong enhancers exerted the most significant influence on the expression of the α-globin gene, while the removal of weak enhancers did not result in any discernible changes [39]. The existence of both strong and weak enhancers has been demonstrated in various studies, including research by Honnell et al. [48] on retinal super-enhancers. These modules have been shown to function additively. Therefore, mice with a deletion of the R0-37 region exhibited microphthalmia, while mice with a deletion of R1-28 had normal eye size but reduced retinal thickness, approximately half that of the wild type. Furthermore, mice with a double deletion of R0-37 and R1-28 suffered from both microphthalmia and retinal thinning. In addition to morphological observations, additional studies, including RNA-seq, were employed to confirm these findings. Consequently, it becomes evident that some super-enhancers are composed of both strong and weak enhancer constituents.

Thus far, researchers have typically employed an approach involving the removal of SE constituents one by one or in various combinations, leaving the remainder of the SE intact [47,48]. However, a distinct approach to SE dissection has unveiled a new paradigm, challenging the prevailing notion that weak enhancers have limited significance [49].

Blayney et al. took a different approach to SE dissection to mitigate potential biases arising from crosstalk among the remaining enhancers [49]. The authors began by completely removing the entire SE and then reconstructed it by adding enhancers back. Their primary objective was to verify the findings of Hay et al. regarding the α-globin SE, which had identified only two SE constituents, R1 and R2, as strong enhancers responsible for 90% of the overall expression of the α-globin gene. Their null hypothesis posited that an SE composed solely of R1 or R2 would result in a 40% or 50% increase in α-globin expression, respectively, as Hay’s group article had shown.

Surprisingly, the results showed that mESCs with R2-only SE had α-globin expression levels fivefold lower than predicted. Mice models with R2-only SE exhibited significantly reduced α-globin expression and were not viable. Similarly, mESCs with R1-only or R1-R2-only SEs failed to provide sufficient gene expression. Elements R3, Rm, and R4 displayed limited gene-activation capacity, consistent with the findings of Hay et al. Interestingly, the addition of any of these elements to R1 or R2 significantly increased α-globin gene expression. Blayney et al. identified R3, Rm, and R4 as facilitators in this process [49].

Hay et al. hypothesised that SEs might exhibit a modular structure. In other words, certain SE constituents may be activated or deactivated during the transitions between different developmental stages. This idea was based on the discovery that some SE elements do not exhibit signs of activity within their natural chromosomal environment [47]. This hypothesis gained support in the study by Bell et al. [50]. For their experiments, they selected ESCs and EpiSC cell lines as models for different developmental stages. In vitro experiments were conducted using these cell models, and in vivo experiments utilised murine embryos at various gestational stages. Bell et al. established a connection between the CpG methylation status of SE constituents and their activity dynamics during cellular differentiation [50].

Throughout the process of development, SE constituents can undergo different outcomes, including:All SE constituents remain active (hypomethylated), and the expression of the target gene remains constant. These genes remain active in both naive and primed cells;All constituent enhancers acquire CpG methylation, resulting in the decommissioning of the SE and the cessation of gene expression associated with the SE. This scenario is typical for SEs that control genes related to maintaining a naive state;Some SE constituents acquire CpG methylation and become inactive, leading to a rearrangement of the SE and a reduction in the expression of the target gene. This allows for fine-tuning of gene expression;All or part of the SE constituents undergo CpG demethylation and become active, enhancing the expression of the target gene. These genes are typically associated with the primed state.

This hypothesis found confirmation and expansion in an upper-mentioned study [48], where the cell-specific active status of SE modules was affirmed in various retinal cell types in mice. For instance, the modular structure of *Vsx2*-SE was demonstrated, with different modules being found to be active in distinct retina cell types. Specifically, two SE constituents, R0-37 and R1-28, were found to influence the proliferation of retinal progenitor cells, while the third module, R3-17, affected the formation of retinal bipolar cells. Interestingly, Honnell et al. regarded evolutionarily conserved SE regions that displayed cell-specific activity as “modules”.

The mechanisms underlying the selection of SE constituents for methylation during the transition from a naive to a primed state remain unclear. It has been speculated that one of the key factors in this selection process might be TFs, which protect SE constituents from methylation. Indeed, those SE constituents that maintained their hypomethylated, active status typically exhibited higher TF occupancy, as well as the presence of the TET1 enzyme, known for its ability to induce CpG demethylation [50].

Thus, the results of the reviewed studies offer evidence for the intrinsic heterogeneity of SEs. Based on specific enhancer signatures, the concentration of SE constituents can be classified into hub and non-hub enhancers, as well as weak and strong enhancers. Hub enhancers are believed to be responsible for the structural organisation of SEs, while strong and weak enhancers contribute to fine-tuning gene expression during cellular differentiation and commitment processes. Additionally, SE constituents can also be categorised as active or poised, in line with the modular SE structure hypothesis.

### 2.4. SE Elements Interaction and seRNA

The question of whether a super-enhancer exhibits greater functionality than the sum of its constituent parts remains open for investigation [25,36,51]. Numerous studies have yielded conflicting results, demonstrating additive, synergetic, and redundant interactions among SE constituents. Additive effects occur when individual SE elements act independently, resulting in a linear increase in target gene expression. Synergetic effects, on the other hand, indicate interdependence among enhancer constituents, such that the deletion of any constituent leads to a significant, nonlinear decrease in gene expression. When manipulations of SE elements do not result in changes in protein production, it is referred to as redundancy.

Experimental evidence supporting additive effects has been found for various SEs, including the extensively studied α-globin SE [47] and the SE of the hormone-dependent gene Wap [25,36,52]. Notably, SE constituents contribute unequally to the expression of the target gene [25,49,52]. To provide further insights, we will focus on the study of α-globin SE, as it yielded similar results.

In the study by Hay et al. [47], sequential and combinatorial removal of α-globin SE elements in mouse models resulted in physiological changes. Heterozygous mice with deleted strong SE constituents remained viable but exhibited worsened blood parameters, such as reduced MCH and MCV concentrations and increased reticulocytes concentrations, characteristic of thalassemia. Homozygous mice, on the other hand, were not viable and died around E14.5. These mice were smaller and paler than wild-type mice, with α-globin RNA levels below 10% of normal levels, likely due to the activity of other (or weaker) enhancer elements.

None of the aforementioned studies observed a purely additive effect; instead, a partially additive and partially redundant effect was observed. Dukler et al. addressed this inconsistency in terminology and proposed a more stringent mathematical approach for classifying interaction types [51]. To validate this theory, the authors re-evaluated the data obtained by Shin et al. for *Wap*-SE [52] and Hay et al. for α-globin SE [47], using different fitting models, including linear (simple linear, linear exponential, and linear logistic) models with and without interactions between enhancer constituents. In both cases, the linear-logarithmic model adequately described the experimental data. Although models allowing for interactions between enhancer constituents demonstrated slightly higher likelihoods, the Bayesian information criterion (BIC) used to assess goodness of fit did not show a significant improvement compared to linear models. Therefore, the authors concluded that there was no evidence of synergy or significant interaction between SE constituents for both SEs, suggesting an additive relationship instead.

High concentrations of Pol2 at super-enhancer regions indicate their active transcription state. Indeed, SEs can be transcribed, giving rise to noncoding super-enhancer RNAs (seRNAs). The concept of seRNA overlaps with the previously introduced concept of enhancer RNA (eRNA) [53]. Notably, it has been demonstrated that eRNAs interact with chromatin remodelling complexes, such as the CREB binding protein (CBP) and PRC2, facilitating H3K27 acetylation and preventing methylation, thereby maintaining an open chromatin state in promoter and enhancer regions [54]. Moreover, eRNAs are implicated in releasing Pol2 from the paused state, promoting its transition to elongation by interacting with positive and negative elongation factor complexes: P-TEFb and NELF [55].

Additionally, evidence suggests (s)eRNA involvement in various molecular processes [56], including retaining transcription factors on enhancers [57], stabilising BRD4 interactions with acetylated histones [58], interacting with the mediator complex [59], and participating in chromatin loop formation during enhancer–promoter interactions [60], including the recruitment of cohesin to oestrogen-regulated enhancers [61].

Super-enhancer RNA synthesis is associated with two types of transcription observed: unidirectional and bidirectional [62]. This results in the production of sense and antisense seRNAs, respectively, which differ in terms of their lifespan and length. Those seRNAs, which are synthesised unidirectionally, are lengthy and more stable, undergoing capping, polyadenylation, and splicing processes [63].

A recent genome-wide study employed a novel approach to investigate the various types of interactions between super-enhancer constituents [64]. The authors analysed the correlation between the total seRNA, assumed to be a proxy for the enhancer activity of an SE, and the mRNA of its target gene by fitting different models (additive, synergistic, and logarithmic) to the data. Additional analyses were conducted to select the best model and validate the results. This genome-wide analysis focused on a dynamic system involving B-cells stimulated to transform into macrophage-like cells. The study revealed that 45% of enhancers cooperated additively, 17% exhibited synergistic interactions, and the remaining interactions were redundant. Enhancer clusters demonstrating synergy were regulated by cell-type-specific TFs and were associated with genes that determine cell identity, such as B-cell-related *BCL7A*, *LEF1*, and *TLE1* and macrophage-related *CITED2* and *LYZ*. Interestingly, no SEs were found to exhibit enhancer cluster cooperation in a synergistic manner. However, this observation can be attributed to the distance limitation imposed by the definition of SEs.

Furthermore, several hypotheses have been proposed regarding the role of SE transcription. Pefanis et al. suggested that the synthesis of sense and antisense seRNA may play a crucial role in SE regulation, forming an R-loop (RNA–DNA hybrids) and recruiting the RNA exosome, which may contribute to the termination of enhancer-associated transcription throughout the genome [65].

Gurumurthy, Bungert, et al. proposed a Pol II transfer model, suggesting that transcription plays a pivotal role in SE function [66]. According to this model, multiple enhancer elements are associated with LCRs and SEs, and accordingly, SEs themselves act as transcriptional complex assemblers, released through transcription termination and transcriptional cleavage. Through a transient looping mechanism (i.e., enhancer–promoter contact formation by loop generation and phase separation), elongation-competent transcription complexes are relocated to the target genes’ promoters.

This model explains the observed burst-like (unstable and frequently aborted) transcription pattern and the additive properties of SEs, as each enhancer carries an equal number of transcriptional complexes. Premature transcription termination is a characteristic feature of SEs, regulated by the integrator and/or WD-domain containing protein 82 [67]. In 2021, the authors published experimental data for β-globin supporting the Pol II transfer model [68].

In conclusion, it is evident that there is no consistent interaction pattern among SE elements. Different studies have provided evidence of additive, synergistic, and redundant interactions between SE constituents. Among these, partial additivity (partial redundancy) was found to be the most prevalent.

Several hypotheses have been proposed to explain the phenomenon of partial redundancy. For instance, redundancy may be necessary for fine-tuning gene expression or ensuring robustness in gene expression. Thus, the regulation of SE constituent activity may contribute to gene fine-tuning in response to various cellular stimuli, such as developmental processes [69]. Additionally, redundancy within SEs may facilitate stable gene expression in the event of the loss of an active enhancer due to mutations [70,71].

However, in addition to partial additive effects, some authors have observed synergistic interactions among elements in certain SEs. According to Cramer et al. [64], this synergistic interaction may enable rapid switching on and off of genes during processes like cell differentiation. Nonetheless, due to the limited number of studies in this area, further data are required to substantiate these conjectures.

### 2.5. SE Search Methods

Computational methods for a genome-wide enhancer and, consequently, super-enhancer prediction can be grouped into three main strategies. The first strategy involves a motif-based approach, which analyses the frequency of existing transcription-factor binding sites (TFBS) across the genome. While this method is widely used and useful for initial analysis, its predictive power can be weak due to differences between predicted and in vivo TFBS distributions [72,73].

To compensate for these limitations, various probabilistic models have been developed, including hidden Markov models (HMMs) [74] and k-mer approaches [75]. These methods take into account the sequence context of the analysed genomic region and are designed to minimise the high false-positive predictions associated with older motif-based algorithms. However, both approaches have their disadvantages, including limited specificity in predicting super-enhancers in vertebrate genomes and inaccuracies in predicting short and dense enhancer subclasses.

The third strategy for enhancer prediction is based on SE evolutionary sequence conservation. This approach may reach a relatively high rate of success in some species (e.g., three pluripotency-associated SEs in *Placentalia*: with up to 80% validation rates in reporter assays [76], indicating SE importance in regulating gene expression. However, this method is restricted to a small subgroup of the general population of potential super-enhancers, limiting its applicability.

One popular non-ML tool for super-enhancer prediction is ROSE (rank ordering of super-enhancers). It was created as a computational framework for discerning the genomic regions that exhibit transcriptional activity, taking into account the ChIP-Seq signals distribution and expression regulation of multiple genes.

ROSE was adapted for super-enhancers ranking using ChIP-Seq data to identify the candidate SE regions that have high levels of histone acetylation (H3K27ac) [7,77,78]. It utilises a statistical approach to rank the candidate regions, employing their enrichment in active enhancer markers, which can be varied through the implementation of diverse ChipSeq experiments (e.g., using also MC and BRD4 data). The ordered list of these regions is then evaluated to identify the top-ranking regions that are likely to be super-enhancers.

ROSE has proven to be a useful tool in identifying super-enhancers across different cell types and species [79,80,81]. Additionally, ROSE has been extensively used in cancer research to identify the key regulatory elements that drive the expression of oncogenes and tumour-suppressor genes [82,83]. Its application has also extended to developmental biology, where it has been used to identify the enhancers that control the identity and function of different cell lineages [84,85].

In summary, the ROSE algorithm’s ability to quickly identify the genomic regions that drive the expression of one or multiple genes makes it a valuable asset in understanding the SE transcriptional regulation of diverse cell types and diseases. On the other hand, ROSE has limitations in predictive power, as soon as the histone modifications spectrum is limited to a few methylation or acetylation marks. Apart from that, ROSE does not imply either topological data or sequence information, which may be crucial for studying SE constituents’ functions.

There are several methods for super-enhancer prediction that involve machine-learning (ML) algorithms. Supervised and semisupervised learning are commonly used to train on selected marks, which are represented by sets of known super-enhancers and non-SE elements. Specifically, supervised models include two types of approaches: discriminative and generative.

Discriminative models directly estimate the posterior probabilities and aim to maximise the separative power between SEs and non-SEs using specified features with various optimisation techniques (e.g., piEnPred [86]). In contrast, generative models try to model underlying probability distributions and use feature data encoding to deliver a probabilistic separation for both classes of elements (e.g., CHROMATIX [87]).

Some machine-learning algorithms, such as random forests [88], RNN [89], ANN [90], CNN [91], and SVMs [92], have demonstrated resistance to overfitting. However, even ensemble methods may not achieve prediction accuracy due to training-data limitations, such as small feature sets and low generalisation power.

Unsupervised methods can be used instead when forming a training set or providing data for supervised approaches is not possible. These methods separate the genome into short contigs, which are labelled as a class if they have similar features (such as ENCODE and ChromHMM [93]). Unsupervised model resolution can be increased with higher frequency genome binning. Unsupervised methods have shown acceptable predictive power and could be used for the prediction of regulatory elements, such as promoters, enhancers and super-enhancers, across genomes [94].

In summary, each of these computational methods for SE prediction has its strengths and weaknesses. A combination of methods utilising motif-based approaches for initial analysis followed by probabilistic models or evolutionary sequence conservation may prove to be a more efficient approach for super-enhancer prediction. In the meantime, machine-learning algorithms are particularly useful in the prediction of SE sequences. While supervised and semisupervised learning methods show promising results, limitations in training data can limit their effectiveness. Unsupervised methods might be a useful alternative in these cases, providing acceptable predictive power for super-enhancer prediction.

### 2.6. SE Databases

Since the inception of research in the field of super-enhancers, a substantial volume of data has been accumulated, thereby necessitating the establishment of a standardised storage system. Presently, to the best of our knowledge, there are four SE databases (db) accessible: dbSUPER [95], SEdb [96], SEA [97,98], and SEanalysis [99,100].

In-depth details regarding each of these databases, as well as a comprehensive comparison between them, are provided in Table 2. This table presents a comparative analysis of various characteristics of the databases, encompassing organism diversity, SE search pipelines, SE annotation, SE visualisation, analysis tools, and links with external resources, among others.

In all of the examined databases, SE coordinates were determined using a standardised pipeline. Specifically, researchers accessed ChIP-seq data for specific SE markers (e.g., H3K27ac) from publicly available sources. They subsequently performed read alignment using the Bowtie tool, peak calling using MACS, and SE identification via the ROSE algorithm.

Data were collected for both the human genome and select model organisms. SEA v.3 demonstrated remarkable organism diversity, including data for Homo sapiens and 10 other organisms, such as *M. musculus*, *D. melanogaster*, *C. elegans*, and zebrafish, among others. In contrast, other db were primarily focused on human and mouse SEs.

Apart from interspecies diversity, SEs also exhibit cell-type specificity, thus emphasising the importance of annotating SEs in different cell lines for novel insights. This aspect is particularly highlighted in SEdb v.2.0, the largest SE database for humans and mice, which contains information on approximately 1167518 and 550226 SEs, respectively. Notably, SEanalysis 2.0 utilises data from SEdb v.2.0, making it the most comprehensive source of SE information for human and mouse organisms. Furthermore, SEdb v.2.0 is actively maintained and received its latest update in 2022, unlike dbSUPER, an earlier SE database that ceased updates in 2017.

In addition to SE coordinates, the majority of SE databases (SEdb v.2.0, SEA v.3.0, and SEanalysis) also incorporate further data to facilitate SE analysis. These include SE-associated genes, both predicted and experimentally validated transcription-factor binding sites, single nucleotide polymorphisms (SNPs), CRISPR–Cas9 target sites, and more.

Furthermore, SEA v.3.0 and SEanalysis offer pathway analysis to elucidate the biological processes involving SEs. Additionally, the SE databases are interconnected with external analytical tools such as GREAT (dbSUPER, SEdb v.2.0, SEanalysis 2.0) and Enrichr (SEA v.3.0). Some databases also support data export to the GALAXY platform (dbSUPER, SEA v.3.0).

Summing it all up, the existing super-enhancer databases serve as valuable repositories of information for SE research. Their focus on discrepancies in SE location between organisms (SEA v.3.0) and cell lines in humans and mice (SEdb v.2.0, dbSUPER), as well as their incorporation of pathway analysis (SEanalysis), provide a diverse range of data analyses. This enables researchers to gain a holistic understanding and obtain various data for novel investigations, including the identification of potential therapeutic targets for various diseases, including cancer, where SE dysregulation has been implicated in pathogenesis.

### 2.7. SE Classification Attempts and Research Context

#### 2.7.1. SE Classification

To enhance our comprehension of the functional properties and potential therapeutic uses of super-enhancers, various endeavours have been made to categorise these entities. In this discourse, we present a synthesis of insights regarding these classification endeavours.

In a previous analysis conducted by our team [88], we propose a potential classification scheme for super-enhancers, originally suggested by Young et al. [7] in 2013. The identification of these regulatory elements was achieved through an investigation of the distribution pattern of the H3K27ac histone modification, which is closely associated with active enhancers and promoters. Notably, super-enhancers were found to exhibit larger dimensions and greater transcriptional activity compared to typical enhancers. Furthermore, they exhibited a remarkable enrichment in proximity to genes that play crucial roles in determining cell identity and disease development. Consequently, the SEs can be classified in a combinatorial manner based on the data acquired during experimental investigations.

In a seminal work by Jung, Ryu et al. [80], the discovery of a novel subtype of SEs with a noncanonical functional role was elucidated. Through a comprehensive investigation of SE specificity across diverse cell lines, the authors categorised SEs into three distinct classes:Unique SEs—these SEs conform to the conventional definition of SEs and exhibit cell-type-specific gene regulatory functions;Nonunique SEs—this class of SEs is present in tissues that share a common function. For instance, the TJP3 (tight junction protein 3) gene, responsible for maintaining junctional integrity in intestinal cells of the digestive system organ tissues (such as the stomach, sigmoid colon, and small intestine), falls under this category;“Common” SEs—these SEs exhibit a strong association with genes that are universally highly expressed and are remarkably conserved throughout the human genome. Integrative analysis of 3D chromatin loops revealed that cell-type common SEs play a crucial role in the formation and rapid restoration of chromatin loops. Notably, loops enriched with common SEs displayed a 12-fold faster recovery rate compared to loops enriched with unique SEs [80].

The proposed classification methodology offers a comprehensive perspective, shedding light on the intricate interplay of temporal and abundance factors that influence cellular fate. Notably, the identification of tissue-specific SE clusters, typically observed during the later stages of the differentiation process, aligns with the categorisation of the so-called de novo super-enhancers [85].

This temporal classification approach serves to enhance the classification efficacy initially introduced by Jung et al. [80], thus contributing to the formation of a more comprehensive classification framework. However, to deepen our comprehension of super-enhancers and their functional dynamics across various stages of cellular development, further investigation into the structural and functional attributes of SEs is imperative. SE classification advances will undoubtedly enhance our understanding of the intricate mechanisms underlying SE function.

#### 2.7.2. LCR and SE Intersection

Super-enhancers (SEs) and locus-control regions (LCRs) are both cis-regulatory elements involved in the regulation of gene expression.

LCRs are specific DNA regions that act as regulatory elements involved in ensuring precise spatiotemporal control of gene expression. They are responsible for coordinating the expression of multiple genes in a given genomic region, often associated with the development and differentiation of cells. LCRs are involved in long-range interactions with their target genes and play a crucial role in maintaining gene-expression patterns during development. Also, SEs are often associated with cell identity and have been implicated in the control of tissue-specific and developmental genes. The presence of SEs has been linked to increased gene expression and the maintenance of cell identity.

While SEs and LCRs are both involved in the regulation of gene expression [67], their relationship and potential overlap remain enigmatic, and our understanding of gene regulation is constantly changing with the development of genomics. Some researchers have long been fascinated by the relationship between locus-control regions (LCRs) and super-enhancers (SEs) in orchestrating gene expression. In an attempt to unravel this complexity, simple in silico experimental approaches (BedTools, BedSect, etc.) offer various options to explore the intersection of LCR and SE coordinates within the human genome and can provide valuable information about LCR and SE functional concordance.

Our understanding of gene regulation is constantly changing with the development of genomics. Some researchers have long been fascinated by the intricate relationship between locus-control regions (LCRs) and super-enhancers (SEs) in orchestrating gene expression. In pursuit of unravelling this complexity, in silico experimental approaches (BedTools, BedSect, etc.) offer various options to explore the intersection of LCR and SE coordinates within the human genome.

Our aim was to ascertain whether these two distinct regulatory elements exhibit any consistent patterns of intersection and possible historical and functional inheritance. Using data obtained from the NCBI and SEdb databases, we generated bed files for LCRs and SEs, respectively, and performed coordinates intersection via the BedTools intersect.

However, a closer examination reveals an absence of regularity in this intersection, highlighting the enigmatic nature of the relationship and the challenges in interpreting the results (Figure 4). The absence of a consistent pattern raises intriguing questions about the nature of the LCR and SE relationship. Are these elements truly independent entities, merely coinciding in the vast genomic landscape? Or do they possess an underlying functional connection that eludes our current understanding?

To fully appreciate the implications of this irregular intersection, it is crucial to delve into the functions attributed to LCRs and SEs. LCRs are recognised as vital regulators of gene expression, involved in ensuring precise spatiotemporal control of genes during development and cellular differentiation. On the other hand, SEs have emerged as key players in driving high-level gene expression, often associated with cell identity and disease progression. While the functions of LCRs and SEs seem distinct on the surface, their intersection challenges this dichotomy. The irregularity of their coexistence highlights the intricate web of genomic regulation, where regulatory elements work in concert, often defying our preconceived notions.

In conclusion, our in silico experiment has shed light on the enigmatic intersection between LCR and SE coordinates within the human genome. The absence of intersection regularity emphasises the randomness of these encounters, making the interpretation of their functional significance challenging. It is evident that LCR and SE coordinates may intersect in the human genome but do not necessarily correlate considering their described functions. This realisation calls for further investigations into the multifaceted nature of gene regulation and the interplay of regulatory elements within the genome.

#### 2.7.3. Evolutionary Conservation of SEs

Super-enhancers are believed to control the expression of genes critical to cellular identity, differentiation, and disease-related pathways. However, SE conservation across different cell types and tissues remains unclear.

Conservation of super-enhancers has been observed in mammals, as demonstrated by a study comparing ChIP-seq H3K27ac peaks in adipose, skeletal muscle, brain, liver, lung, and spleen tissues of humans, mice, and pigs. The results showed that 8–16% of pig SEs are functionally conserved, despite no sequence conservation being observed [101].

Moreover, highly conserved SE has been found in *Mammalia Placentalia*, regulating pluripotency genes such as *SOX2*, *PIM1*, and *FGFR1*. Interestingly, while TF binding sites were conserved, species-specific differences in their density could lead to varied SE activity and pluripotency transcription programs [102].

In contrast, no highly conserved SEs were discovered when analysing SEs between vertebrates such as humans, mice, and zebrafish. However, SEs regulating orthologous genes were, on average, more conserved than SEs of nonorthologous genes [13].

Further research investigating SE dynamics during cardiomyocyte differentiation showed conserved temporal patterns of SE in humans and mice. The study found that the proportion of de novo established, decommissioned, and temporary hierarchical SE in humans and mice was similar [103]. The conservation of SEs across mammal species suggests that these regulatory elements have putative functional features that have been preserved throughout evolution.

## 3. Super-Enhancers and Diseases

The super-enhancer concept is partially based on the observation that cell-identity maintenance in mESCs is highly sensitive to the inhibition of the transcriptional coactivator mediator and cohesin complex, as well as to the perturbation of master TFs [9].

Similar scenarios in multiple cancer types have also been frequently associated with alterations in the expression of oncogenes (genes associated with the initiation of cell division and apoptosis, the control of whose expression prevents the uncontrolled cell division characteristic of cancer cells), oncogenic TFs, the production of chimeric TFs, mutations in transcriptional regulators, and SE-dependent altered functional crosstalk.

The mentioned aberrant SE-driven transcriptional programs could be involved in both cancer development and the progression of some other diseases, including certain immune diseases, neurodegenerative diseases, and diabetes [104,105], which will be discussed further in detail.

### 3.1. Oncology

According to the hallmarks of cancer, oncologies rely on specific gene (oncogenes) activity to maintain proliferative signalling and initiate metastasis. In malignant cells, oncogenes are amplified or overexpressed, and this overexpression is typically regulated at the transcriptional level. Many cis-regulatory elements and trans-acting factors work together to control gene expression. One of the reasons for the oncogenic expression increase may be pathological binding to SE, with the formation of the so-called oncogenic SE. Alterations in SE activity in cancer involve multiple mechanisms, leading to dysregulation of transcriptional regulators and SE-associated genomic abnormalities, such as:chromosomal rearrangements leading to a convergence of SE with a specific oncogene;(super-)enhancer hijacking;SE focal amplifications;insertions, deletions, and other types of mutations in DNA that form binding sites for new TFs, leading to the formation of SE;insertions, deletions, and other types of mutations in the SE that disrupt TFs binding sites, leading to disruption of the SE function;deactivation of SE associated with some tumour-suppressor genes;disruption of TAD and sub-TAD boundaries;TFs increase in expression and overexpression, leading to pathological gene activation;viral oncogenes (Epstein–Barr and HPV viruses);some types of chemotherapy (e.g., cisplatin);other pathognomonic events.

We summarised in the Appendix A Table A1 the diversity of cancer types and their subclassification, focusing on the anomalous interactions of SE with the oncogenes identified in a number of neoplasms, the SEs and TFs said to interact with them, the percentage of cases with discovered alterations, possible molecular pathogenetic ways, downstream-effected associated pathways, and SE possible inhibiting effectors.

Studies of some cancer types have identified mechanisms that could potentially be common to a number of other malignancies that share similar characteristics. One such example is the MYC gene, whose overexpression is responsible for many pathways important for tumourigenesis in several oncologic diseases, namely leukaemia, lung cancer, endometrial cancer, prostate cancer, glioma, and pancreatic ductal adenocarcinoma [99,106,107,108,109,110].

Oncological diseases of close localisations often have common SE-related mechanisms of onset. Thus, haematological oncologies such as high-grade serous ovarian cancer (HGSOC), epithelial ovarian cancer (EOC) and other ovarian cancer subtypes have several SE-associated pathological mechanisms, namely BRD4 and SE copy-number amplification similar to breast cancer [111]; ovarian cancer-like shared mechanisms of cisplatin-induced SE activation (concluded based on elevated seRNA expression); and BRD4 amplification. Of note, HDAC1/3/7 inhibition can result in the inhibition of the cancer stem cell (CSC) phenotype by downregulating multiple SE-associated oncogenes in breast cancer stem cells [112].

Alternative molecular mechanisms of SE-driven transcriptional programs in cancer cells may represent cell-type/tissue-specific TFs or oncogenes/master transcription factors of so-called core regulatory circuitry (CRC’, alternatively, core regulatory network/circuits) overexpression. The concept of CRC’ was introduced in 2005 when Boyer et al. showed that the transcription factors OCT4, SOX2, and NANOG are core TFs that co-occupy promoters of target genes and control a pluripotency program in embryonic stem cells (ESCs) [113].

And later, in 2013, when these transcription factors were also revealed to co-occupy super-enhancers of genes associated with pluripotency and each other, this concept was clarified and adjusted. Importantly, core master TFs not only stand at the head of the hierarchy of regulatory networks and drive other TFs’ activation but also bind to each other’s super-enhancers, thereby forming an interconnected autoregulatory loop [114].

One of the most popular CRC’ oncogenes in cancer is the *CD47* gene—encoding a transmembrane protein which partners with integrins and binds membrane regulatory proteins—caused by aberrant SE constituents and helps cancer cells escape phagocytosis in B-cell lymphoma. Similar mechanisms can take place in other types of cancer, such as breast and subtypes of lymphoblastic leukaemia [115].

The main classes of cell metabolic pathways that are damaged or drastically changed during tumourigenesis and oncological transformation (with the help of SE areas and their influence on the level of expression of target genes) are as follows: cell growth, proliferation, and self-renewal processes; cell survival (including viability and clonogenic capacity); antiapoptotic functionality; and tumour immune escape (evasion of immunosurveillance).

When talking about the exact affected biochemical pathways, examples include the notch signalling pathway affected by NOTCH1 mutations in T-ALL, the PI3K-AKT signalling pathway influenced by PIK3CA mutations in breast and colon cancer, and the Wnt/β-catenin pathway in CRC’ and LIHC.

Overall, Table A1 provides a comprehensive overview of TFs and genes implicated in various cancer nosologies and subtypes, identifies common genes, including TFs and other transcriptional regulators affected/associated by specific SEs, and highlights the molecular mechanisms and pathways known by date.

It is important to clarify that we refer to the overall dynamics of SEs—including the distribution of SEs and their regulatory mechanisms—by synthesizing various research perspectives on how individual structural elements may influence SE functionality. Our analysis seeks to identify both convergences and divergences in these perspectives, forming a basis for proposing a hierarchy of SE functionality (both functional and structural). Regarding the methodology for predicting SE dynamics, our current work does not provide specific methods but rather sets the stage for future in-depth in silico analyses aimed at exploring the conservation and constitutive nature of SEs and their elements.

Our further discussion will be based on the Table A1 data. We first study the alterations that are mainly associated with changes in single SEs or SE constituents (such as chromosomal rearrangements, single-nucleotide polymorphisms, somatic mutations, and viral infection-mediated changes). Afterwards, we move on to discussing more complex mechanisms involving groups of super-enhancers through finer regulation of TF complexes, including posttranslational modifications and seRNAs.

#### 3.1.1. Chromosomal Rearrangements and SE Dynamics

SEs, acquired or lost due to chromosomal rearrangements, were extensively identified in haematological malignancies, namely t(3;8)(q26;q24) AML [108], inv(3)/t(3;3) AML [116,117] (Figure 5), AML with MLL rearrangements [118], multiple myeloma, and, in one solid cancer, fusion-positive rhabdomyosarcoma [119]. All these rearrangements have previously been known to be associated with a high risk of cancer development, but the discovery of their connection with super-enhancer dysregulation sheds light on major or additional mechanisms underlying malignisation.

Chromosomal rearrangements leading to the convergence of *EphA2*-SE with a specific oncogene *EphA2* might be another common mechanism to boost PI3K/AKT and Wnt/β-catenin pathways and be shared between several oncological nosologies, namely colorectal cancer, breast cancer, cervical cancer, etc. [120].

SEs mediating EVI1 oncogene overexpression have been revealed in AMLs harbouring different rearrangements. Relocation of the GATA2 enhancer element from the 3q21 to the 3q26.2 loci results in *EVI1*-SE formation and *GATA2*-SE loss in inv(3)/t(3;3) AML, leading to EVI1 upregulation and haploinsufficiency of GATA2, respectively. Primitive hematopoietic precursors are sensitive to GATA2 homeostasis disturbance; thus, reduced GATA2 expression in the right spatiotemporal context can promote cell malignisation via EVI1 activation [117].

A later study revealed that MYC-SE translocates to the *EVI1 locus*, and one of the SE modules regulates EVI1 overexpression in t(3;8)(q26;q24) AMLs. SE-promoter looping is facilitated by multiple CTCF-binding sites in hijacked MYC-SE and CTCF-binding sites upstream of EVI1 TSS [108]. Since the CTCF binding site upstream of the EVI1 promoter was preserved in AML harbouring 3q21 rearrangements and seemed to be essential to SE hijacking, it has been proposed that the mechanism providing SE-promoter interaction is common for all 3q26-rearranged AMLs demonstrating EVI1 overexpression.

Frequent translocation t(4;14)(p16.3;q32.3) in multiple myeloma (MM) leads to transcriptional activation by a distal SE (induced by the histone lysine methyltransferase NSD2) of the histone chaperone HJURP. HJURP overexpression confers aggressive behaviour in t(4;14)-positive, making HJURP a valuable therapeutic target in patients with t(4;14)+ MM [121].

Translocation (6;8)(p21;q24) in blastic plasmacytoid dendritic cell neoplasm leads to the association of the super-enhancer of RUNX2 and the MYC promoter [122]. Table A1 also highlights specific rearrangements and mutations linked to SEs in different cancers; for instance, gene fusions like *BCR-ABL1* and *PML-RARA* are often seen in chronic myeloid leukaemia (CML) and acute promyelocytic leukaemia (APL), respectively.

A translocation in Ewing sarcoma, the second most common malignant bone tumour, most commonly between chromosomes 11 and 22, often results in a fusion between the 5′ region of the *EWS* gene and the 3′ region of *FLI1*. The fusion (chimeric) protein, EWS/FLI1, has DNA-binding activity and promotes expression. It is a known oncogene. The EWS/FLI1 occupancy of super-enhancers is considerably higher compared to typical enhancers [123]. Ewing sarcoma onset is often due to a chimeric oncoprotein EWS-FLI1. SE-associated transcripts are significantly enriched in EWS-FLI1 target genes (e.g., *MEIS* and *APCDD1*), contribute to the aberrant transcriptional network of the disease, and mediate the exceptional sensitivity to transcriptional inhibition [124].

#### 3.1.2. SNPs, Somatic Mutations, and SE Dynamics

SNPs located in noncoding regions may contribute to SE formation or loss via TF’s preference to bind to a specific allele over an alternative. SNP-caused SE dysregulation associated with cancer development has been observed in colorectal cancer (CRC), chronic lymphocytic leukaemia (CLL), and neuroblastoma.

Patients with inflammatory bowel disease (IBD) often carry another risk SNP, the rs2836754 allele (T), which leads to disease-specific *ETS2*-SE activation both in IBD and CRC since TF MECOM prefers binding at the T allele over C in this region. ETS2 downstream genes regulate the inflammatory response; thus, ETS2 SE-driven overexpression results in permanent inflammation and may serve as a key mechanism, increasing the predisposition to the development of these diseases or the progression from IBD to CRC, as IBD is a disease with a high risk for CRC development [125] (Figure 6).

Reduced *BMF*-SE activity in chronic lymphocytic leukaemia (CLL) due to carrying the rs539846 risk allele (A) at the RELA binding site leads to the disruption of RELA binding, decreased BMF expression, and, therefore, BMF’s inability to inhibit the antiapoptotic protein BCL2. In this study, no association between SNP rs539846 and patient survival has been observed, which can indicate the importance of BMF downregulation in the early stages of CLL. However, the analysis had <50% power [126].

SNP rs2168101 G>T has been shown to be specifically associated with neuroblastoma and to be located at the GATA3 binding site within the *LMO1*-SE region. Among 127 neuroblastoma primary tumours, 80% carried the G/G phenotype (G/G = 102, G/T = 25, T = 0), and LMO1 mRNA expression was significantly decreased in G/T tumours compared to G/G tumours [127]. Importantly, neuroblastoma cases with rs2168101 = G/G exhibited significantly worse event-free survival and overall survival compared to rs2168101 = G/T or T/T cases in the European-American cohort (n = 2101). Based on the reviewed articles, we suggest that SNP-caused SE loss or formation is primarily associated with the risk of cancer development; however, harbouring a homozygous risk allele genotype may be crucial for survival and disease outcome, as was observed for the rs2168101 = G/G genotype in neuroblastoma.

In addition to SNPs, somatic mutations occurring in SE regions may also contribute to cancer development. In T-cell acute lymphoblastic leukaemia (T-ALL) lacking TAL1^d^ abnormality or TAL1 chromosomal rearrangements, heterozygous indel mutations in the region 7.5 kb upstream of TAL1 TSS result in de novo MYB binding motif (or even two motifs as in Jurkat cells) and thereby the emergence of *TAL1*-SE. MYB has been shown to be crucial for *TAL1*-SE activation because it recruits the CBP protein, which opens the chromatin and allows other members of the TAL1 complex to bind at *TAL1*-SE. *TAL1*-SE-mediated monoallelic overexpression of the *TAL1* oncogene leads to the establishment of the T-ALL cell state, which suggests that targeting TAL1 expression or its downstream targets may be a potential therapeutic strategy for treating T-ALL [128].

Additionally, Asnafi et al. suggest that, within a specific oncogene-driven cancer, the mechanism of oncogene dysregulation itself can identify clinically distinct patient subgroups and pave the way for future SE-targeting therapy. Therefore, T-ALL patients harbouring 5′SE mutations of the *TAL1* oncogene represent a specific subgroup with poor prognosis, irrespective of the level of oncogene dysregulation [103].

#### 3.1.3. Viral Infection-Mediated Changes and SE Dynamics

SEs dysregulation through different mechanisms has been revealed in HPV-related cervical cancer [129,130,131,132], developing on the background of human papillomavirus (HPV) integration (Figure 7).

Dooley, Katharine E et al. have identified Brd4-dependent SE, formed de novo by tandem copies of HPV16 DNA integrated into the genome of cervical neoplasia-derived cell line W12 subclone 20861. De novo formed SE regulates *E6/E7* viral oncogene overexpression [130]. Further, they have revealed that HPV16 integrates into an intergenic region located at chromosome 2 and flanked by a DNA sequence containing a basal cellular enhancer. Then, the viral genome was coamplified in about 26 copies with a flanking cellular locus of size ~25 kb. The synergy of amplificated viral upstream regulatory regions and adjacent hijacked cellular enhancers results in the formation of SE-like elements, mediating overexpression of the viral oncogenes *E6* and *E7* and neighbouring cellular genes, particularly *FOSL1* [131]. Prominent nuclear Brd4 focus has not been observed via immunofluorescence analysis in the other two W12 subclones, 20831 and 20862, and in an additional six cervical carcinoma-derived cell lines.

An integrative analysis of multiomics data from six HPV-positive and three HPV-negative cell lines investigated the genome-wide transcriptional impact of HPV integration using HPV integration detection, SE identification, SE-associated gene expression, and extrachromosomal DNA (ecDNA). Seven high-ranking cellular SEs caused by HPV integration—the so-called HPV breakpoint-induced cellular SEs (BP-cSEs)—were discovered, which resulted in the intra- and interchromosomal set of gene dysregulation. Among them are *ISCA1*, *SLC25A28*, *ISCA2*, *GLRX5*, *FANCC*, *ING5*, *SCO2*, *RFC2*, and some other genes linked to cancer-related pathways [129].

In another study, HPV *E6* has been confirmed to interact with histone H3K4me2/3-specific demethylase KDM5C via the central and C-terminal portions of the protein and promote KDM5C degradation in a proteasome-dependent manner [132].

Reduced H3K4me3 and increased H3K4me1 have been observed at *EGFR* and *c-MET* super-enhancers in CaSki cells with restored KDM5C expression compared to CaSki control cells, which is consistent with previous findings, evidencing that KDM5C occupies almost all active super-enhancers and regulates dynamic H3K4me1–H3K4me3 interchange [133].

Thus, HPV *E6* induces *EGFR* and *c-MET* proto-oncogenes SE-mediated expression via KDMC5C destabilisation, leading to SE activation. Since this mechanism is not directly related to HPV integration into the host-cell genome, KDMC5 downregulation and the consequent H3K4me1–H3K4me3 interchange may be involved in SE dysregulation in other cancer types, which is also suggested in the mentioned study [133].

Another study shows enhancer reprogramming through TRIM11 interaction with KDMC5. TRIM11 is highly expressed and KDM5C is expressed lower in breast cancer patient tissues, and their expressions are negatively correlated [134].

In the EBV-transformed lymphoblastoid cell line (LCL), widely used as a model of EBV-associated B-lymphomas, all four oncogenic EBV-proteins, EBNAs, and all five NF-kB subunits were revealed to co-occupy approximately 1800 enhancer sites, including 187 sites identified as super-enhancers. According to the proposed model, cell NF-kB, being activated by EBV-derived protein LMP1, and EBV EBNAs co-occupy enhancer sites and recruit other cell TFs, such as EBF, a major pioneering B-cell lineage factor increasing chromatin accessibility, STAT5, and NFAT, and thereby create so-called EBV super-enhancers (EBV SEs). EBV SEs regulate key B-cell growth and survival genes, including *MYC* and *BCL2* [135].

In the study of the nasopharyngeal carcinoma (NPC) global histone epigenetic landscape provided by the same group, NF-κB motifs were found to be enriched in NPC-specific peaks, revealed from the H3K27ac ChIP-seq signal, but not in control-specific peaks. NPC pathogenesis can be associated with EBV infection; thus, in EBV-positive NPC, NF-κB may be activated by viral proteins LMP1, as in B-lymphomas or LMP2, and more consistently expressed in NPC [136]. NF-kB activation via EBV proteins apparently plays one of the key roles in enhancer gain and epigenetic landscape alterations in EBV-related cancers.

As it was mentioned, there are some examples of SE dynamics alteration mechanisms similarity between different viral species, as is the case between HPV and EBV. At the same time, variations of such mechanisms within the same HPV species can also be found. Thus, viral infections represent a distinct class of molecular pathological changes that affect the dynamics of super-enhancers. Further study is required to establish the full variety of mechanisms associated with viral integration into the host genome and, as a consequence, the altered dynamics of super-enhancers.

#### 3.1.4. Global SE Alterations

Many authors identify so-called global changes in super-enhancer dynamics or global SE alterations when talking about widespread changes in the profile of SE activity studied within genome-wide approaches.

In breast cancer, changes in the super-enhancer landscape have been found at different stages of breast-tumour progression. Ropri et al. have compared SE landscapes in progression series generated from the MCF10A cell line, which included normal MCF10A cells, premalignant cells forming atypical ductal hyperplasia (MCF10AT1), MCF10AT1 xenograft-derived cells forming ductal carcinoma in situ (MCF1DSIC), and MCF10AT1 xenograft-derived cells forming poorly differentiated malignant tumours (MCF1CA1). When compared to normal MCF10A cells, 383/173, 684/12, and 28/259 SEs were newly acquired/lost at the AT1, DCIS, and CA1 stages, respectively. Gene ontology assessment to SE closest genes revealed many acquired pathways, including STAT signalling in AT1, NF-kB and FOXA2 signalling in DCIS, and CD147 in cancer-cell motility in CA1 cells. Further comparison analysis of SEs in patient BC samples showed *EphA2*-SE, acquired at the DSIC stage, was found in 34/47 ER+ and 10/10 TNBC samples. Interestingly, *EphA2*-SE was not detected in TNBC cell lines. Thus, TNBC and ER+ samples had common SEs arising apparently at the DSIC stage, which may serve as markers of potential DSIC transformation to invasive ductal carcinoma [137]. *TFAP2A*-SE, acquired in the CA1 cells, was uniquely found in all TNBC samples and in the 10/11 TNBC cell lines, probably indicating differences between TN and ER+ breast cancers. Other more significant findings are described in Table A1.

Genome-wide reprogramming of the enhancer and SE landscape was found in gastric adenocarcinoma (GC). Utilising Nano-ChIPseq for H3K27ac, H3K4me3, and H3K4me1 histone modifications, Ooi, Wen Fong, et al. generated 110 chromatin profiles from 19 primary GCs, 19 matched normal gastric tissues, and 11 GC cell lines. Since cell lines are exclusively epithelial in nature, GC lines were used to discover GC-associated enhancers. Considering all H3 modification signals, researchers detected 36,973 distal enhancers in all GC cell lines, of which 47% were found in at least 2 GC cell lines out of 11 and were considered recurrent, indicating high variability of enhancer activity across cell lines. Interestingly, SEs demonstrated a significantly greater tendency to be recurrent when compared to typical enhancers; across 3759 nonredundant predicted SEs identified via the ROSE algorithm, 3345 turned out to be recurrent. Predicted SE was associated with known oncogenes, such as *MYC*, *KLF5*, and *MALAT1* [138].

Then, the activity of cell-line-predicted enhancers was compared between GC tumour samples and matched normal gastric tissues. Based on exhibited alteration in activity between cancer and a normal state, SEs statuses in GC were classified as somatic gain (n = 1748, 47%), somatic loss (18%), unaltered (n = 416, 11%), or inactive (n = 808, 21%). The authors suggested that regions corresponding to SEs associated with somatic loss may represent regions epigenetically silenced in primary tumours but reactivated in cell lines during in vitro culture. Unaltered SEs were associated with regions related to general tissue functions or housekeeping genes. Interestingly, inactive SEs exhibited low recurrence in GC cell lines. Importantly, further mapping of the catalogues of disease-associated SNPs reported from 1470 GWAS against recurrent SEs revealed that somatically altered super-enhancers were enriched in genetic risk SNPs associated with cancer and inflammatory gastrointestinal disease (a disease with a high predisposition to GC). This was not observed for unaltered SEs, which provides evidence about the possible contribution of SNPs located in the SE region to SE gain or loss in GC. It is important to note that, when comparing recurrently gained SE in primary GC samples with an intestinal histotype (n = 10) and samples with a diffuse histotype (n = 6), 471 and 224 SEs were revealed as intestinal-specific and diffuse-specific, respectively. Common for both histotypes were 516 predicted SEs [138].

Genome-wide profiles of H3K27ac marks were generated for 20 samples of primary epithelial ovarian cancers (EOC) taken from patients with clear-cell ovarian cancer (CCOC), endometrioid ovarian cancer (EnOC), high-grade serous ovarian cancer (HGSOC), and mucinous ovarian cancer (MOC). Each of the EOC subtypes was represented by five samples. Across SEs revealed in HGSOC tissues, 93 SEs were associated with genes encoding lncRNAs, of which only 3, UCA1, SNHG9, and SNHG15, were correlated with prognosis. *UCA1*-SE was also found in three other EOC subtypes, but not in normal precursor cells: fallopian tube secretory epithelial cells (FTSECs) or ovarian surface epithelial cells (OSECs). *UCA1* depletion via the CRISPR–Cas9 system revealed that UCA1 is important for cell growth, and further experiments showed the role of UCA1 in the regulation of the Hippo-YAP signalling pathway. Thus, it is speculated that SE-mediated UCA1 overexpression drives ovarian cancer development [139].

It is important to note that in the described work, researchers focused their attention exclusively on lncRNA-SEs because the major aim of the study was to detect lncRNAs associated with EOC and reveal the regulatory mechanisms of those lncRNAs. However, based on the GEO page corresponding to this experiment, it may be concluded that histotype-specific regions of H3K27ac signals were associated with enhancers when common peaks were predominantly located at promoters. Interestingly, only 20 histotype-specific peaks were identified in EnOC, whereas in other histotypes, the number of unique peaks amounts to several thousand. Moreover, in the matched RNA-sequence dataset, only 16 DEGs were specific for EnOC, whereas for other histotypes, several hundred DEGs were identified [140].

A colorectal cancer genome-wide ChIP-Seq analysis of 73 pairs of CRC tissues in total was found to contain 5590 gain and 1100 loss variant enhancer sites, as well as 334 gain and 121 lost variant super-enhancer loci. Via motif and core regulatory circuitry analysis, multiple important transcription factors in colorectal cancer were predicted, with PHF19, TBC1D16 and KLF3 having a special role to play in colorectal cancer carcinogenesis [141].

Noel, Pawan et al. have demonstrated that the SE profile in pancreatic ductal adenocarcinoma (PDAC) epithelial cells differs from the SE profile in cancer-associated fibroblasts (CAFs) cell lines isolated from patients with PDAC. Super-enhancer regions in PDAC cancer-cell lines were associated with such genes as POLR2E, PARK7, and MYC and were enriched in GO biological processes related to transcription regulation, apoptosis, and immune function, while SE-associated genes in CAFs (e.g., *COL1A1*, *COL1A2*, *TGFBI*) were enriched in extracellular matrix organisation, angiogenesis, and hypoxia processes. The important result of the study is the therapeutic relevance of triptolide in targeting both tumour epithelial cells and CAFs. Triptolide mediates disruption of cell-specific SEs and suppresses SEs-associated gene transcription via inhibiting XBP subunits of the TFIIH complex. The authors speculated that triptolide treatment can reprogram tumour-supporting crosstalk between cells in the PDAC microenvironment through disruption of SE-mediated transcriptional programs both in tumour cells and in CAFs [142].

Significant alteration of the SE landscape in cancer, when compared to normal tissue, has been detected in human hepatocellular carcinoma. SE acquired in HCC was associated with a lot of known oncogenes, such as *YAP1*, *CCND1*, *E2F2*, *EGFR*, and *MYC*. The expression profile of HCC SE genes obviously separated HCC samples from normal tissue samples in the unsupervised clustering analysis [143].

Changes in the regulatory regions of the *PAX5 locus* are critical in the development of chronic lymphocytic leukaemia (CLL). PAX5, being the key regulator of SE in CLL, modifies the SE landscape and the whole core regulatory circuitry [144].

The triple-negative subtype of breast cancer (TNBC) is a heterogeneous disease lacking known molecular drivers with significantly poorer survival rates compared to other breast cancer subtypes. TNBC has been demonstrated to be one of the breast cancer subtypes with a critical role in SE-dependent onset. It has been shown that over 2500 unique SEs are acquired by tumour cells, pseudo-receptor *BAMBI* has been identified as a SE-associated gene [101] and *ANLN*, *FOXC1*, and *MET* genes as TNBC-specific genes regulated by SEs [102]. High levels of SE-associated NSMCE2 strongly correlate with patients’ poor response to chemotherapy, especially for patients diagnosed with aggressive TNBC and HER2+ breast cancer types, *MAL2* is the second novel SE-associated gene identified in [104].

Lung adenocarcinoma, characterised by focal amplification of SE [105], is associated with more than 781 abnormal activated SEs [145]. Similar genetic profiles between lung, head and neck, and cervical cancers might suggest the conservation of SE regions between these diseases and tissues [146,147].

Furthermore, in ovarian cancer, SE genome-wide changes have been observed after repeated cisplatin treatment. Cell-type-specific transcription-factor ISL1, an SE regulator in ovarian cancer cells, apparently is the main driver of SE plasticity induced by cisplatin treatment. ISL1 is being suppressed in cisplatin-treated cells, which reorganises the global SE program and mediates the adaptation process in the cells undergoing near-to-death experience [148].

Moreover, specific hormone stimulation may influence SE dynamics as well. For instance, Dex treatment in breast cancer resulted in genome-wide reorganisation of the enhancer and SE landscape. A Dex-specific SE encompasses DDIT4 and four glucocorticoid receptor (GR) binding sites allowing for a loop-switching mechanism to induce DDIT4 transcription under Dex treatment [149].

Another humoral regulation mechanism is known in prostate cancer. Androgen receptors in prostate cancer repress MYC. This repression is independent of AR chromatin binding and is driven by coactivator redistribution. AR causes disruption of the interaction between the *MYC*-SE within the *PCAT1* gene and the *MYC* promoter. Androgen deprivation increases MYC expression [150].

#### 3.1.5. Critical Proteins of SE Regulation Complex Overexpression

Overexpression of trans-acting components of the SE machinery has been discovered in a number of cancers and, probably, additionally contributes to the gain of SE-mediated transcription. So, CDK7, MED1, EP300, and BRD4 were overexpressed in primary HCC samples in contrast with normal tissue samples [143]. Interestingly, mutations and deletions in TP53, a gene encoding the p53 protein guarding genome stability, were significantly enriched in HCC patients with CDK7, BRD4, EP300, and MED1 overexpression, which indicates a possible association between TP53 sequence alterations and upregulation of SE complex trans-acting components.

Transcriptional cyclin-dependent kinases (CDKs), particularly CDK7, facilitate transcription initiation and promote productive elongation through phosphorylation of the RNAPII c-terminal domain (CTD). CDK7 upregulation was revealed in 35 osteosarcoma primary tumours compared to normal tissue. All 10 examined osteosarcoma cell lines have shown high sensitivity to THZ2, a selective CDK7 inhibitor [151]. In thyroid carcinoma, CDK7 expression was significantly increased in anaplastic thyroid carcinoma (ATC) samples compared to papillary thyroid cancer (PTC) samples, and cells with the ATC subtype were sensitive to THZ1 treatment [152].

In colorectal cancer, MC38 sublines with a higher expression of CDK12, another regulator of elongation, demonstrated enhanced metastatic competence in Western blotting experiments. *DCBLD2*, *NTSR1*, *CAV1*, *CCDC137*, *CDC25B*, *PRKACB*, and SE-associated genes were also overexpressed in cancer compared to the norm, were sensitive to a low dose of SR-4835, a selective inhibitor of CDK12, and showed expression reductions greater than 1.5-fold after SR-4835 treatment. *CCDC137*, whose function in colorectal cancer remains unknown, has been identified as an oncogene promoting proliferation and stemness. *CCDC137* depletion significantly reduced migration and invasion of cells occurring against the background of CDK12 overexpression, which probably indicates CCDC137’s important role in liver metastasis promotion [153].

Ovarian cancer patients have the highest overall expression of BRD4 and the highest rate of genetic amplifications at the *BRD4 locus* across the entirety of the TCGA Pan-Cancer dataset; ~11% of patients with ovarian cancer have an amplification of the region containing the *BRD4* gene. Importantly, amplification of SE-associated genomic regions is significantly frequent compared to ovarian cancer genome-wide amplification in CNV analysis of ~600 patients with ovarian cancer. Thus, BRD4 amplification probably serves as additional fuel, contributing to SE-mediated transcription alterations primarily associated with SE CNVs. Moreover, about 6% of patients with uterine corpus endometrial carcinoma, and also skin cutaneous melanoma, have mutations in the *BRD4* gene [111].

Importantly, BRD4 has been revealed to play a role in the suppression of SE activity. In breast cancer, BRD4 reorganises chromatin and facilitates recruitment of the LSD1–NuRD complex to SEs, which in turn with BRD4 occupies SEs, thereby repressing genes, and is functionally linked to drug resistance. JQ1 long-time treatment declines the LSD1 protein level and decommissions the BRD4–LSD1–NuRD complex, inducing resistance to JQ1 and a spectrum of other compounds. Thus, such results indicate the dual role of BRD4 in regulating the activity of super-enhancers and also highlight the possible negative effects caused by JQ1 treatment [154].

#### 3.1.6. TFs Induce SE Activation and Regulate SE-Mediated Programs

In a number of cancers, the gain or loss of SE activity occurs due to the alteration of TF dynamics, which bind to corresponding SE regions and drive SE-mediated transcription. The dysregulation of some TFs may specifically affect just one or several genes, but global SE-activity alterations, particularly due to the dysregulation of master TFs, seem to occur more often in cancer. Thus, the *MYC* oncogene was revealed to be upregulated and responsible for strong SE-mediated overexpression of other critical oncogenes such as *CDK6* and *TGFB2* [155] in osteosarcoma. Additionally, MYC gene overexpression is responsible for many pathways important for tumourigenesis in several oncologic diseases: leukaemia, lung cancer, endometrial cancer, prostate cancer, glioma, and pancreatic ductal adenocarcinoma [105,107,108,109,110,118,150].

Other frequently occurring genes include *TP53* (as well as *TP73*, a *TP53* family SE-associated gene present in adult T-cell leukaemia/lymphoma [156] and *TP63*), *AR* [157], *ERG*, *ETS* (*ETV1*, *ETV4* [158], and *ETV6*) and *IRF* (*IRF1*, *IRF2*, *IRF4* [159], and *IRF8*) family members, etc. Notably, TFs, like ERG, androgen receptor (AR), and ETV1, are commonly observed in prostate cancer, while TP53 and MYC are frequently involved in multiple cancer types.

Another study addresses the distinct gene signature of lung fibroblasts, according to which multiple key TFs of lung mesenchyme development include TBX2, TBX4, and TBX5 (T-box TFs) and are associated with SEs. These TFs are downregulated and hypermethylated in lung cancer-associated fibroblasts (CAFs), suggesting an association between epigenetic silencing of these factors and phenotypic alteration of lung fibroblasts in cancer. Yet, TBX4 is functionally active not only during organogenesis but also in the cellular homeostasis of lung fibroblasts. This supports the notion that fibroblasts retain “positional memory” [160].

Many additional instances will be expounded upon subsequently, within the framework of elucidating the factors influencing the altered dynamics of transcription factors (TFs) and within the context of unravelling the mechanisms that establish oncogenic transcriptional programs.

#### 3.1.7. CRCs and Autoregulatory Loops in TFs-Modulated SE Landscape

In a series of works, master TFs have been shown to form autoregulatory loops, i.e., master TFs occupying genome-wide super-enhancers also occupy their own SEs. Thus, TFs directly establish positive feedback loops and maintain or enhance the transcriptional programmes they drive.

In T-ALL, mentioned above, indels in regions upstream of TAL1 TSS lead to de novo MYB binding-site creation, and then CBP, recruited by MYB, provides chromatin accessibility and makes possible the binding of the TAL1 complex at de novo SE. The TAL1 complex includes TAL1 itself, GATA3, and RUNX1. These TFs have previously been revealed to form an interconnected autoregulatory loop and bind at the *MYB* enhancer [161]. Since MYB and TAL1 co-occupy approximately 80% of TAL1 binding sites throughout the whole genome and form positive interconnected autoregulatory loops via binding to each other’s enhancers, it can be concluded that MYB together with the TAL1 complex serve as core regulatory circuitry in T-ALL. However, it is worth noting that MYB is the key factor in this CRC’, as MYB is crucial for *TAL1*-SE establishment and, consequently, the initiation of autoregulatory positive feedback circuitry and the stabilisation of the TAL1-regulated oncogenic program. The key role of MYB was confirmed by *MYB* knockdown via siRNA, which turned out to be sufficient for the depletion of TAL1, RUNX3, and GATA3 [128].

Alternative examples of autoregulation have been found for the following master TFs: FOSL1 in HNCC [162], MYC in colorectal cancer [163], ASCL1 in small lung adenocarcinoma [164], ISL1 in ovarian cancer [148], FOXC1 in triple-negative breast cancer [102], and MYCN in MYCN-amplified neuroblastoma [165]. In addition, TFs can enhance transcriptional programmes indirectly, e.g., ERα regulates SE-driven transcription of the RET gene, which, in turn, activates the RET/RAS/RAF/MEK/ERK/p90RSK/ERα phosphorylation cascade, thereby promoting ERα activity and, consequently, ERα-regulated gene transcription [166].

Mesenchymal tumours obtain their own characteristic factors. Fusion-positive rhabdomyosarcoma (FP-RMS)—RMS with t(2;13)(q35;q14) chromosomal translocation resulting in *PAX3-FOXO1* fusion gene formation—seems to be regulated by CRC’, formed by PAX3-FOXO1, MYOD1, MYCN, and MYOG. PAX3-FOXO1 protein serves as a pioneer factor, opening chromatin and recruiting BRD4, and thereby induces the formation of de novo SEs. PAX3-FOXO1 sets up interconnected autoregulatory loops via binding to its own SE and to SEs of myogenic master TFs, namely MYOD1 and MYCN, which, in turn, establish *MYOG*-SE, and together with MYOG and PAX3-FOXO1 co-bind *PAX3-FOXO1*-SE, *MYOD*-SE, and *MYCN*-SE. It is important to note that myogenic master TFs bind almost all revealed SEs, while PAX3-FOXO1 occupies just 47% of them. Thus, the pioneering factor PAX3-FOXO1 is the main driver of SE-mediated transcription reprogramming, but it needs further myogenic master TF activity to maintain widespread transcription regulation of genes responsible for RMS cell identity [167]. Another example of the CRC’ critical component in osteosarcoma is LIF being an essential factor under the control of osteosarcoma-specific SE. Its expression positively correlates with the stem-cell core factor genes in osteosarcoma [168].

Jiang et al. identified TP63, SOX2, and KLF5 as the master TFs responsible for orchestrating the CRC’ in ESCC cells through the establishment and maintenance of chromatin accessibility and binding to SEs regions. In this study, it was shown that the knockdown of at least one of the core TFs leads to disruption of the whole regulatory program. Thus, the authors believe that the TP63, SOX2, and KLF5 trios occupy their own SE elements as well as each other’s, thereby forming interconnected autoregulatory loops. In their previous studies, researchers also found some SCC-specific lncRNAs regulated by SCC-specific SEs, namely CCAT1 and LINC01503 [169]. Further, CCAT1 was identified as a key SE-regulated lncRNA in three other SCC types. Interestingly, CCAT1-SE was shown to be co-occupied by TP63 and SOX2, while *LINC01503*-SE was only bound by TP63, precisely by the TP63 isoform ΔNp63.

In lung adenocarcinoma (LUAD), master TFs ELF3, EHF, and TGIF1 have been revealed to form CRC associated with widespread alterations of SEs. The loss-of-function assay demonstrated that each of the master TFs is essential for LUAD cell survival, invasion, and metastasis [170].

The group 3 subtype medulloblastoma (G3-MB), the most common malignant paediatric brain tumour, CRC’ comprises the 14 SE-associated genes (so-called vital SE-associated genes, vSE) which include the three common TFs (MYC, OTX2, and CRX) and 11 newly identified downstream effector genes, including novel SE-associated genes, *PSMB5* and *ARL4D* (subtype specific). The conserved SE-associated oncogenic signature between primary tumour lines and tissues of G3-MB is enriched with subtype-specific upregulated tumour-dependent genes, and MB patients with such an enrichment exhibit a worse prognosis [171].

In the case of other haematological malignancies, the TCF3-HLF chimeric TF in paediatric acute lymphoblastic leukaemia (ALL) and the ETO2-GLIS2 chimeric TF in paediatric acute megakaryoblastic leukaemia (AMKL) activate MYC expression, which, in turn, accumulates in the SEs of malignant cells, leading to a conserved MYC-driven type of transformation programme [172]. The majority of SEs in haematologic cancer cells are generated by key oncogenic drivers and are associated with genes that maintain hematopoietic identity (such as *IKAROS* and *STAT5*) [44].

Interestingly, some paediatric tumours were found to differ in the SE landscape [173,174]. Moreover, in infant neuroblastoma 2, major SE-directed molecular subtypes have been described, namely the ADRN and MES subtypes. The ISX drug has been shown to reprogramme SE activity and switch NB cells from an ADRN subtype towards a growth-retarded MES-like state, sharing strong transcriptional overlap with GN, a benign and highly differentiated tumour of the neural crest, opening new insights into cancer therapy [175].

Thus, the clustering of various tumour subtypes based on SE profiling makes it possible to enhance the existing clinical tumour classification, which indicates the important role of SEs in the formation of a certain tumour phenotype from the point of view of its localisation and age of onset.

#### 3.1.8. Sample Mechanism of TF-Mediated Pro-SE Looping: The Role of YY1

Binding sites in SEs regions were found for Yin-Yang 1 (YY1), another TF associated with cancer progression, hepatocellular carcinoma, high-grade serous ovarian cancer, and triple-negative breast cancer [102,111,176]. YY1 was shown to bind SEs driving QKI, RAE1, FOXC1, and MET1 transcription in HCC, HGSOC, and TNBC, respectively, which positively affects cancer-cell migration and invasion (Table A1). Furthermore, Han, Jingxia, et al. proposed the mechanism of YY1-activated QKI overexpression during HCC, according to which YY1 forms complexes with p65 and p300, inducing DNA loop formation and bringing *QKI*-SE and *QKI* promoter closer.

In a little more detail, YY1 binds to SE and the promoter of *QK1*, while p65 binds to the *QK1* promoter, and p300 serves as a mediator stabilising the YY1–p65–p300 complex [176]. Huang et al. also predicted YY1 binding sites in the *FOXC1* promoter in addition to *FOXC1*-associated SE utilising ENCODE [102]. Perhaps a similar mechanism, described for QKI, may be observed in other types of cancer.

It is worth noting that Weintraub, Abraham S., et al. previously showed that YY1 structurally regulates enhancer–promoter interactions in a manner similar to CTCF-mediated DNA looping in humans and mice. YY1 and CTCF are ubiquitously expressed in mammalian cells and, based on the authors’ results, are required for normal gene transcription through enhancer–promoter looping, which apparently is a general feature for gene control in mammalian cells. The authors demonstrated that YY1 occupancy is observed genome-wide in both typical enhancers and super-enhancers and is likely to be cell type specific, as human ChIP-seq data exhibited patterns of YY1 binding differing between lymphoblastoid cells, colorectal cancer cells, hepatocellular carcinoma cells, embryonic stem cells, T-ALL cells, and CML cells [177]. Thus, one can assume that YY1 binding to SEs of cancer-related genes contributes to gene dysregulation during cancer development and progression via facilitating SE–promoter interactions.

#### 3.1.9. Copy Number Variation and Overexpression of TFs Leading to SEs Alteration

One of the possible reasons for TF dysregulation is genomic amplification of the gene encoding the respective TF, which further leads to transcriptional amplification. For example, genomically amplified MYCN has been shown to drive global transcriptional amplification of active genes through binding with their promoters and enhancers in MYCN-amplified neuroblastoma. Moreover, THZ1 induces apoptosis of MYCN-amplified cells and correlates with the downregulation of SE-associated genes, unlike MYCN-non-amplified cells, which evidences the selective vulnerability of MYCN-amplified neuroblastoma cells to this inhibitor. Interestingly, after the recalculation of the MYCN enhancer rank for one copy of MYCN, the MYCN enhancer was still identified as SE and had a strong H3K27Ac signal. Based on this, it was suggested that the main contribution to the higher signal in MYCN-amplified neuroblastoma cells compared to MYCN-non-amplified neuroblastoma cells is not due to an increased copy number of the super-enhancer sequence [165]. However, enhancer elements of SE may act in a synergistic manner and, thus, contribute to an increased signal from each other. Thus, we assume that this result can be discussed.

Another example is the genomic amplification of TP63 and SOX2, master regulators that establish and maintain the global SE landscape in oesophageal SCC, HNSCC, and LSCC [178].

However, additional copies of a gene are not always the cause of TF dysregulation, and often the mechanism of dysregulation is unclear. Zhang et al. revealed FOSL1 as the master regulator in HNSCC, promoting tumourigenesis and metastasis via establishing SEs associated with key oncogenes, such as *SNAI2*, *CD44*, and *FOSL1* itself. SNAI2 is another TF regulating the epithelial–mesenchymal transition (EMT). In this study, FOSL1 was significantly upregulated in HNSCC tumour tissues and correlated with metastasis of HNSCC, but the mechanism of upregulation has not been investigated [162].

#### 3.1.10. Spliced Isoforms of TFs Regulating SEs

Another example of altered TF activity and lifespan is the protein isoforms undergoing alternative splicing. For instance, ΔNP63α is a prominent isoform of TP63, driven by SEs associated with basal cell-specific genes in nasopharyngeal cancer (NPC) cells. NPC originates via malignant transformation of the pseudostratified nasopharyngeal epithelium, composed of basal and luminal cells. Basal cell-specific proteins are highly expressed, whereas luminal cell proteins are downregulated in NPC, implying a perturbation of basal-to-luminal differentiation during NPC development, a process associated with distinct SE landscapes. ΔNP63α is a master factor contributing to the perturbation of luminal differentiation [179].

ΔNp63 was also studied as the squamous cell carcinoma (SCC)-specific oncogenic factor, regulating some SE-associated lncRNAs (seRNAs). It was found that ΔNp63 binds to *LINC015030*-SE and contributes to its SCC-associated transcription activation, suggesting that LINC015030 is an important downstream effector of TP63 [178].

#### 3.1.11. Post-Translational Modifications Changing TFs Affinity

Nguyen, Duy T et al. have discovered that K13-acetylated (acK13)-HOXB13 is a crucial regulator of SE selectivity before the development of castrate-resistant prostate cancer (CRPC). HOXB13, a lineage-specific pioneer TF, is specifically targeted by CBP/p300, histone acetyltransferase, to mediate K13-acetylation of HOXB13, which transforms HOXB13 into a pro-CRPC TF. (acK13)-HOXB13 differentially interacts with chromatin remodelling proteins, bromodomain-containing proteins, and CTCF, when compared to unmodified HOXB13.

Moreover, the acK13-HOXB13 binding motif at the SE region differed from the unmodified protein, and the most common acK13-HOXB13 motifs in PC were enriched with motifs of TF associated with the epithelial–mesenchymal transition, such as FOXA1, ZEB1, and FOXO1. A ChIP-seq of (acK13)-HOXB13, HOXB13, and H3K27ac signals analysis in tumour and normal samples has revealed that (acK13)-HOXB13 occupies an SE targeting lineage (*AR*, *HOXB13*), CRPC promoting (*ACK1*), prostate cancer diagnostic (*FOLH1*, *SPON2*), and angiogenesis-related (*VEGFA*) genes in cancer samples.

It has been demonstrated that increased expression of ACK1, one of the most remarkable SE-associated genes, overrides the loss of androgen stimulation, and human prostate tumour organoids expressing HOXB13 showed significant resistance to AR antagonists but were sensitive to (R)-9b, an ACK1 selective inhibitor [180].

#### 3.1.12. seRNA (eRNA)

eRNAs are noncoding RNAs, transcribed from enhancers. Oncogenic SEs generate noncoding SE RNAs (seRNAs) that exert a critical function in malignancy through the powerful regulation of target gene expression. Although the significance of SE transcription is not fully understood, some seRNAs have been shown to have specific functions, and some are associated with oncological and autoimmune diseases [63].

Certain seRNAs can interact with transcription factors (TFs) and cofactors, including CBP proteins responsible for H3K27ac modification and CTCF proteins involved in chromatin loop formation at SE–promoter interactions. seRNA activity has been demonstrated both in cis and in trans and was reviewed by Xiao et al. [181]. Another example is a JUN-mediated seRNA, associated with metastasis (seRNA-*NPCM*), which forms an R-loop to simultaneously regulate distal target (*NDRG1*) and neighbouring (*TRIB1*) genes to promote the metastasis of nasopharyngeal carcinoma (NPC) [182].

eRNA expression is sensitive to BRD4 perturbation, which has been demonstrated in multiple studies utilising JQ1 treatment. Interestingly, SE-driven transcription, in turn, may be sensitive to eRNA expression. eRNA transcribed from the *ALDH1A1*-SE region is critical for SE-driven *ALDH1A1* mRNA transcription, promoting stem-like features in high-grade serous ovarian cancer [183]. Since *ALDH1A1*-eRNA is regulated by BRD4, and both BRD4 inhibition and eRNA knockdown reduce *ALDH1A1* mRNA levels, it is suggested that BRD4 first activates eRNA transcription, and then, eRNA together with BRD4 establishes *ALDH1A1*-SE-driven transcription. Another, newer study has shown that *CHPT1*–seRNA expression in castration-resistant prostate cancer is also BRD4-dependent. Furthermore, authors have experimentally revealed that *CHPT1*-seRNA interacts with BRD4 protein in two BRD4 regions containing evolutionarily conserved lysine-rich motifs through its 349–552-nt region and regulates BRD4 occupancy in *CHPT1*-SE [184]. Thus, *CHPT1*–eRNA–BRD4 interaction plays an important role in SE activation in Enz-resistant CRPC. Probably, eRNAs may play a similar function to that in SE-driven gene dysregulation in other cancers.

In addition to the proposed recruitment of BRD4 to SE regions, eRNA probably influences SE formation in other ways. In the lymphoblastoid cell line (LCL) derived from EBV-transformed B-cells, *MYC*-ESE (EBV SE) eRNA has been essential to *MYC*-ESE and *MYC* TSS looping, which has been confirmed on LCLs transduced with control shRNA or targeting shRNA targeting eRNA utilising 3C qPCR assays [185]. It is worth mentioning that *MYC*-ESE eRNA is sensitive not only to BRD4 inhibition but also to the inactivation of EBV nuclear antigen 2 (EBNA2), which is also crucial to SE formation.

Several studies have shown that SE-driven lncRNAs can play an important role in tumour development. *RP11-569A11.1*, a SE-lncRNA, has tumour suppressor functions by regulating IFIT2. It is significantly downregulated in colorectal cancer [186]. Another example is *LINC01004*, which is significantly upregulated in liver cancer tissues. It is associated with a poor prognosis. It was shown that *LINC01004* promotes the cell proliferation and metastasis of hepatocellular carcinoma [187]. SE-associated lncRNAs in stomach adenocarcinoma are related to immune markers. For example, TM4SF1-AS1 suppresses T-cell-mediated killing and exhibits the immune response to anti-PD1 therapy [188]. *LINC01977*, another SE-associated lncRNA, promotes proliferation and invasion both in vitro and in vivo in lung adenocarcinoma. *LINC01977* interacts with SMAD3 and induces its transport to the nucleus, which facilitates the interaction between SMAD3 and CBP/P300. The SMAD3–CBP–P300 complex activates ZEB1, the central switch of EMT [189]. Moreover, *LINC00162*, transcribed from the SE region in bladder cancer, interacts with THRAP3, thereby disrupting THRAP3’s ability to positively regulate PTTG1IP expression. Since PTTG1IP inhibits cell proliferation and promotes apoptosis, its downregulation due to *LINC00162* overexpression and subsequent *LINC00162*–THRAP3 interaction promotes bladder cancer [190]. For example, the TF HSF1, exerting a multifaceted role in tumourigenesis, specifically activates the SE region in colorectal cancer. HSF1-mediated lncRNA–*LINC00857* promotes cell growth via regulating SLC1A5/ASCT2-mediated glutamine transport, proving the relevance of SE-lncRNA regulation in the SE molecular pathogenesis [191], the idea also supported by [192].

Hepatocellular carcinoma (HCC) apart from focal SE amplification, for instance, has a hybrid mechanism of oncological transformation due to lncRNA-*DAW* acting together with liver-specific SE. Ectopic expression of lncRNA-*DAW* enhances tumour growth both in vivo and in vitro. Wnt2 acts as a downstream effector of lncRNA-*DAW* according to the RNA sequencing results. Another association was shown for lncRNA-*DAW* with EZH2 [193]. EZH2 in HCC is also associated with somatically acquired SE-activated *SIRT7*. Together, they induce cooperative epigenetic silencing, as shown in [194].

The role of SE-regulated lncRNA-*LINC00094* of BRD3OS (named *SERLOC* by the authors) was also shown in cutaneous squamous cell carcinoma (CSCC)—the most common metastatic skin cancer. *SERLOC* was identified as a biomarker for invasion and metastasis of CSCC; its incidence is increasing worldwide and the disease has a poor prognosis [195].

Some studies even highlight the so-called ceRNAs (lncRNA-mediated competing endogenous RNAs). These ceRNAs are identified based on the evaluation of global and local regulatory direction consistency of expression, as is done in the oesophageal squamous cell carcinoma (ESCC) model [196].

Thus, eRNA functions are not limited only to the establishment of SE at the loci they are transcribed from but also include the regulation of the expression of other genes, supporting the idea of cooperative regulation between transcription factors and (s)eRNA. Additionally, SE-associated lncRNAs (seRNAs) can serve as markers of cancer and serve as targets for therapy. Nevertheless, further research is required to confirm these functions of eRNAs.

### 3.2. Inflammation

Inflammation represents a significant aspect of pathophysiology, manifesting in numerous chronic conditions or as an independent disease entity. Extensive research in the fields of infectious diseases and autoimmune disorders has contributed to a deeper understanding of the fundamental mechanisms underlying inflammation. Notably, super-enhancers (SEs) have emerged as key regulatory elements with a pivotal role in orchestrating inflammatory processes and modulating the interplay of various coregulated genetic features.

The transcription factor NF-κB, recognised as a master regulator, plays a crucial role in the activation of inflammatory signalling cascades. NF-κB forms complexes associated with chromatin, thereby facilitating the transcription of proinflammatory factors [197]. Studies have demonstrated that NF-κB is recruited to enhancer regions, contributing to the establishment of chromatin topology and the formation of enhancer clusters in close proximity to target genes [195,198]. Notably, J.D. Brown et al. [199] revealed NF-κB-dependent activation of endothelial enhancers, leading to the formation of proinflammatory super-enhancers (SEs) and subsequent global changes in the BRD4 landscape. This activation resulted in the upregulation of proinflammatory endothelial factors such as TNF-α. Interestingly, under cytokine stimulation, a rapid loss of noninflammatory SEs was observed, suggesting the existence of an immediate reorganisation mechanism essential for the formation of novel proinflammatory SEs [199]. Similar regulatory mechanisms were observed in human adipocytes, where TNF-α treatment induced the acute gain of NF-κB-bound SEs and the loss of basal SEs [200]. Specifically, in mesodermal cell lines such as human endothelial cells, TNF-α proinflammatory signalling led to the activation of KDM7A and UTX enzymes, resulting in their delocalisation from dormant SEs and a subsequent loss of overall adhesive activity in the cells [201].

SEs have been established as critical determinants of cell identity and differentiation in various contexts. For instance, SEs have been strongly associated with the expression of IL-9 and the induction of Th9 cell differentiation. Selective knockdown of Brd4 and Med1 proteins, as well as the use of JQ1 inhibitors, resulted in the downregulation of anti-inflammatory IL-9 expression and the arrest of Th9 cell differentiation, specifically disrupting SE formation. This regulatory mechanism appears to be characteristic of Th9 cells compared to other T-cell lineages and is associated with RelB- and p300-mediated chromatin acetylation [202]. Conversely, most T-cell lineages possess alternative mechanisms for SE folding and activation regulation. For example, SEs associated with juvenile idiopathic arthritis (JIA) exhibit a high enrichment of ETS and RUNX1 binding motifs, which are typically activated by proinflammatory signalling. Cytokine activation leads to alterations in the existing SE landscape and the de novo establishment of proinflammatory SEs. Experimental observations, including JQ1-mediated repression of *JIA* genes and immune-related SEs, indirectly confirmed this effect [203]. In the context of inflammatory bowel disease (IBD), approximately half of the associated SNPs have been identified within the SE-regulated regions of CD4+ T-cells. Furthermore, hierarchical SE regulation of the *BACH2 locus* in Th1, Th2, and Th17 cells has been shown to repress lineage determination in response to cytokine stimuli [204].

In addition to cytokine involvement, various methods have been employed to model inflammatory stimuli and processes. For instance, the treatment of different cell populations with lipopolysaccharide (LPS) is a common technique used to induce acute proinflammatory signalling. Studies utilising macrophages as a model system have demonstrated the upregulation of SE-associated genes involved in cellular metabolism and nuclear organisation following LPS treatment [199]. Furthermore, LPS treatment has a profound impact on the expression of enhancer RNAs (eRNAs), which, in turn, affects the expression of a wide range of genetic elements [205,206,207].

Insights into the role of SEs during tumourigenesis have also been elucidated. Inflammatory signals derived from the tumour microenvironment of colorectal cancer (CRC) have been shown to induce critical changes in the epigenetic landscape, leading to the de novo formation of SEs that drive tumourigenesis. Specifically, the tumour microenvironment triggers the deposition of the H3K27ac modification at the *PDZK1IP1* gene, reshaping the SE landscape of CRC cells. Through a positive feedback mechanism, the newly formed SE stimulates PDZK1IP1 expression, promoting malignancy [208].

Collectively, these studies underscore the complex and crucial role of a cumulative and dynamic pool of SEs as a genomic module that not only maintains cellular identity and differentiation trajectories but also responds to external inflammatory cues.

### 3.3. Other Nosologies

Super-enhancers are a promising frontier in the field of therapeutic interventions, presenting prospective paths for early and urgent treatments across different pathological conditions. To date, super-enhancers have been implicated in a variety of complex diseases, including cardiovascular, endocrine, autoimmune disorders, etc., underscoring the critical need for a comprehensive understanding of their role in disease progression. As such, it is important to research the pathology-associated regulatory functions of SEs to pave the way for routine clinical testing, identify novel associations, and pinpoint prospective targets for therapeutic intervention and preventive medicine.

Research has shown that genetic variations, particularly in regions called super-enhancers (SEs), can influence the risk of developing certain diseases, including cardiovascular conditions. Thus, cardiogenetics, a rapidly growing subspecialty that combines the fields of cardiology and clinical genetics, aims at the investigation of the genetic factors underlying cardiovascular diseases. By focusing on identifying and understanding these genetic factors, cardiogenetics serves to improve the diagnosis, treatment, and prevention strategies for different heart conditions [209].

As described in a review paper by Timothy J. Cashman and Chinmay M. Trivedi, several research groups have reported a connection between SEs and cardiac pathologies [210]. For example, a recent study by VanOudenhove et al. [211] revealed the role of cardiac-specific active enhancers during human heart development. These enhancer regions exhibited elevated activity during embryonic cardiogenesis but were repressed in foetal and adult human hearts. The study also identified a significant enrichment of variants associated with atrial fibrillation in these enhancer regions, suggesting that atrial fibrillation may be a congenital cardiac disease rather than an acquired disease.

Another review paper by Samir Ounzain and Thierry Pedrazzini summarised research findings in the field of cardiogenetics over the past 10 years [212]. This review also refers to the pioneering work of Young’s laboratory, where approximately 400 super-enhancers were identified as specific to the adult left ventricle and showed highly conserved heart-specific traits compared to other transcriptional enhancers [7]. These super-enhancers were associated with key cardiac transcription factors, emphasizing their role as crucial regulatory hubs within the cardiac gene regulatory network.

In studies focusing on cancer-cell types, it has been demonstrated that BET inhibitors, such as JQ1, which target the bromodomain reader protein Brd4, can lead to decreased expression of SE-associated oncogenes. This inhibition has shown promising therapeutic effects in multiple myeloma and pathological cardiac remodelling, including heart failure [78].

Coronary artery disease (CAD) is a multifaceted condition influenced by a combination of genetic and environmental factors. Although the precise role of super-enhancers (SEs) in the pathogenesis of CAD is an active area of research, the direct evidence linking the two is currently limited. Nonetheless, comprehending the pathogenesis of CAD and the regulatory role of super-enhancers in gene expression offers valuable insights into potential associations. It is postulated that super-enhancers may govern the expression of genes involved in fundamental processes relevant to CAD, including lipid metabolism, inflammation, and endothelial function.

A study conducted by Gong et al. in 2018 [213] employed GWAS and ChIP-seq data analysis to identify CAD-associated single nucleotide polymorphisms (SNPs) within loci annotated as super-enhancers. The investigation revealed that 366 SNPs within SE annotations exhibited a regulative association with CAD-related genes such as *ZMIZ1*, *CBFA2T3*, *DIP2B*, *ANAPC15*, *TMEM105*, and *NPRL3*. Subsequent analysis involved the reconstruction of a protein interactome, which demonstrated that the identified SNPs influenced the interplay between CAMK2G and MAPK1. These proteins have been shown to be implicated in CAD pathogenesis [214]. Consequently, it is plausible that perturbed SE activity could potentially modulate the expression of diverse genes and contribute to the pathogenesis of CAD. However, further research is needed to establish direct causal connections.

Altogether, the findings suggest that super-enhancers, particularly those specific to the heart, may play important regulatory roles in cardiovascular biology, including the response to stress and the development of cardiovascular diseases.

There were efforts to link other conditions, such as diabetes, to their super-enhancer nature. Diabetes exhibits a heterogeneous pathogenesis characterised by diverse physiological mechanisms. It is classified into type-1 and type-2 diabetes based on genetic or lifestyle-related factors. Exploring the regulatory effects of super-enhancers (SEs) has laid the foundation for novel studies investigating the association between SEs and diabetes. For instance, researchers have postulated a consistent regulation of PD-L1 expression through direct interaction between SE-derived RNA (seRNA) and BRD4, considering the correlation between specific single-nucleotide polymorphisms (SNPs) in the *PDCD1* (*CD274*) gene and the occurrence of type-1 diabetes (T1D). The BRD4-dependent recruitment of P-TEFb to acetylated chromatin enhances CD274 gene expression by phosphorylating RNA polymerase II, thereby promoting the expression of a potential prodiabetic enzyme variant [63].

Additionally, it has been demonstrated that approximately 19% of noncoding region SNPs associated with type-1 diabetes represent around 1.3% of the Th cell genome and are located within classified SE elements. Given the autoimmune nature of T1D and the subsequent destruction of insulin-producing β cells, it is noteworthy that 13 out of the 76 SNPs linked to type-1 diabetes occur within SEs specific to T-helper cells [17].

Despite the apparent similarities between SEs and stretched enhancers (StEs), several studies have reported correlations between specific SNPs in the pancreatic islet epigenomic landscape and type-2 diabetes (T2D). Fundamental research conducted by Weiping Sun et al. led to the identification of 286 putative functional T2D super-enhancer SNPs [24] that exhibited a strong enrichment in T2D-associated genes encoding enzymes involved in pancreatic β-cell function and glucose metabolism [215].

In the realm of neurodegenerative diseases, Alzheimer’s disease (AD) remains a prominent focus of research for clinicians and neuroscience analytical divisions. However, over the past decade, research related to SEs in AD has not amassed significant datasets. Nevertheless, studies focusing on microglia, neurons, and oligodendrocytes have identified 2954 SEs, with 83% of them being highly abundant around promoters displaying elevated H3K27ac levels. Genome-wide association studies (GWAS) have suggested a connection between disease-risk variants and the aforementioned SEs in predisposing gene variants associated with AD [216]. Another study catalogued 27 SNPs linked to AD, with five of them being located within brain tissue super-enhancers. Notably, two of these SNPs were associated with a mutant SE characterised by a small insertion that regulates the *BIN1* gene, which is strongly associated with Alzheimer’s disease [17,217].

Autoimmune diseases arise as a consequence of intricate interactions between genetic, environmental, and immunological factors. The dysregulation of the immune system instigates the generation of autoantibodies and activation of immune cells, subsequently culminating in tissue damage and eliciting inflammatory responses. Although the precise role of super-enhancers (SEs) in the pathogenesis of autoimmune diseases remains incompletely elucidated, it is postulated that SEs may exert control over the expression of genes associated with pivotal processes pertinent to autoimmune diseases, such as immune-cell activation, cytokine production, and tissue-specific autoantigen expression, as demonstrated in a study focusing on lupus erythematosus [218]. A qualitative review conducted by the authors of this study encompassed a broad spectrum of autoimmune diseases, including but not limited to inflammatory bowel disease, Graves’ disease, and atopic dermatitis. Within this review, the authors accumulated an array of facts, encompassing detailed descriptions of various autoimmune diseases. Furthermore, the review highlighted the identification of single nucleotide polymorphisms (SNPs) detected within super-enhancer loci [219].

## 4. Anti-SE Drugs

SE disruption was shown to be promising for a number of diseases, including cancer [17], autoimmune diseases [203], and neurodegenerative [220] and other SE-associated diseases. This is achieved by targeted inhibition of molecules, high concentrations of which are characteristic of SEs, for example, BET, CDK7, HDAC, etc. Studies in this field were predominantly done for malignant tumours, aiming to find new treatment strategies. Therefore, in this section of the review, our main focus will be on the use of inhibitors in oncology.

To begin with, there is a group of BETi (Table 3) that targets one or both bromodomains (BD1 or/and BD2) of Brd2/Brd3/Brd4/Brdt. Between them, the most studied BET inhibitor is JQ1 [221], which inhibits both bromodomains (BD1 and BD2) of Brd2/Brd3/Brd4/Brdt. It was shown to be effective for a wide range of tumour types (for example, diffuse large B-cell lymphoma [16], ovarian cancer [183], colorectal cancer [200], osteosarcoma [155], breast cancer [222], etc.). Some other BET inhibitors were also shown to be effective, for example iBET in diffuse large B-cell lymphoma [16,223], iBET151 (diffuse large B-cell lymphoma, colorectal cancer, ovarian cancer [224,225,226], OTX-015 in diffuse large B-cell lymphoma and neuroblastoma [16,227,228], volasertib in ovarian cancer [226], BAY 1238097 in melanoma [228], etc. (Table 3). A number of them (iBET151, OTX-015, etc.), just like JQ1, target both bromodomains (BD1 and BD2), while other ones have one main target (BAY 1238097 shows strong binding of BD1 and weaker binding of BD2, while ABBV-744 [229] is a selective inhibitor of the BD2 domain). Volasertib stands aside, exhibiting a dual kinase-bromodomain inhibition (it targets both PLK1 and BD1/BD2). Most of them show submicromolar or lower half maximal inhibitory concentration (IC50) (Table 3). 

While most BET inhibitors have shown good results in preclinical studies, a number of them have failed clinical trials due to their high toxicity. JQ1 has a low oral bioavailability and a short half-life (only 1 h); therefore, it is not applicable in the clinic [230]. Moreover, JQ1 has been reported to be rather toxic in neuronal derivative cells [231]. OTX015 (birabresib/MK-8628) was tested in patients with acute leukaemia [232], lymphoma and multiple myeloma [233], castrate-resistant prostate cancer, NMC, and non-small-cell lung cancer [234], but clinical studies were terminated at phase 2 due to lack of clinical activity and not due to safety reasons (NCT02296476). Clinical trials of BAY1238097 in patients with solid tumours and lymphoma [235] and volasertib in patients with acute myeloid leukaemia (NCT02003573) were terminated at phase 1. ABBV-744 shows minimal toxicity in rats and can be used for prostate cancer treatment [229], and is now in clinical trials in combination with ruxolitinib or navitoclax in adult participants with myelofibrosis (NCT04454658). BI 894999 is in the first phase of clinical trials in patients with neoplasms or NUT carcinoma demonstrated clinical activity, but its haematological toxicity should be decreased using synergistic drug combinations (NCT02516553 [236]). 

Another group of inhibitors targets CDK7, a part of transcription factor II H (TFIIH). CDK7 inhibition was shown to suppress SE-linked gene transcription in a number of tumours [165]. CDK7is THZ1 and THZ2 are the most explored [237,238]. The short half-life of THZ1 in vivo (45 min in mouse plasma) can potentially limit its performance [239]. THZ2 has improved pharmacokinetic features compared with THZ1, with a 5-fold improved half-life in vivo [238]. Some other CDK7 inhibitors, for example, SY-1365 and YKL-5-124, show covalent binding to CDK7 and also have low IC50 and positive effects in ovarian and breast cancer, and neuroblastoma, respectively [240,241] (Table 3). SY-1365 clinical studies are terminated due to business decisions (NCT03134638). Clinical trials of YKL-5-124 have not started yet.
ijms-25-03103-t003_Table 3Table 3BET and CDK7 inhibitors that act on super-enhancers.
AgentTargetIC50Tumour TypeReference(s)BET inhibitorsJQ1BD1 and BD2 of Brd2/Brd3/Brd4/Brdt77 nM/33 nM for BRD4 (1/2)Various types[16,84,183,222,223,242,243,244,245]iBET151BD1 and BD2 of Brd2/Brd3/Brd4/Brdt0.79 μMfor BRD4Diffuse large B-cell lymphoma, colorectal cancer, ovarian cancer[16,224,225,226]OTX-015BD1 and BD2 of Brd2/Brd3/Brd4/Brdt0.449 µMDiffuse large B-cell lymphoma, neuroblastoma[16,228]BAY 1238097Strong binding of BD1, weaker binding of BD2<100 nMMelanoma[228]VolasertibDual kinase-bromodomain inhibitor300 nM/770 nMBD1/BD2Ovarian cancer[226]BI 894999BD1 and BD25 nM/41 nMBD1/BD2Acute myeloid leukaemia[246]ABBV-744Selective inhibitor of the BD2 domain4 nMProstate cancer[229]CDK7 inhibitorsTHZ1Cysteine residue located outside of the canonical kinase domain<200 nMSmall-cell lung cancer, breast cancer, OSCC, ATL, NPC, ovarian cancer, GBM, osteosarcoma, cutaneous melanoma, hepatocellular carcinoma, chordoma, CML[143,155,237,238,247,248,249,250,251,252,253,254,255]THZ2Binding to CDK713.9 nMTNBC, osteosarcoma[237]SY-1365Covalent binding to CDK7369 nMOvarian and breast cancer[240]YKL-5-124Covalent binding to CDK78–60 nMNeuroblastoma[241]


Although some BRD4is and CDK7is have not yet been experimentally shown to affect super-enhancers, there is reason to believe that, by having the same targets as the inhibitors discussed above, they potentially affect SE. Some of such inhibitors are promising, for example, samuraciclib, a CDK7 inhibitor that showed antitumour activity and an acceptable safety profile during clinical trials (NCT03363893) for advanced solid malignancies. BET inhibitors ZEN-3694 (NCT04471974) and ZEN003694 (NCT04986423, NCT05327010, NCT05372640, etc.) are currently in clinical trials.

Histone deacetylases and demethylases are other potential targets for inhibition. Histone deacetylase inhibitors (HDACIs) were shown to be rather effective in rhabdomyosarcoma [256], breast, and ovarian tumours [112]. The selectivity of these inhibitors depends on the HDAC isoform. According to their chemical structure, HDACIs can be divided into three large groups: benzamides, hydroxamic acids, and other inhibitors. Benzamides mainly influence HDAC1/2/3, while hydroxamic acids are less selective, or selective for other classes. (Table 4). As for HDAC inhibitors, a number of them, including romidepsin, vorinostat, Panobinostat, and belinostat are approved by the United States Food and Drug Administration (US FDA) [257]. Entinostat has received ‘breakthrough designation’ status from the US FDA for the management of advanced breast cancer [258]. Thus, HDAC inhibitors are characterised by relatively low toxicity and high efficacy.

It was shown that a combination of inhibitors can show synergy. The complex effect of combined treatment strategies also leads to the reduction of drug resistance. For a number of super-enhancer inhibitors, such strategies have already been tested, leading to a positive effect for some tumours (Table 5).

JQ1 and THZ1 have shown synergy in cell lines and mice for neuroblastoma. In cell lines, it was shown that JQ1 and THZ1 alone reduce the growth of cells, while their combination leads to no growth at all and even the reduction of the cell number. In vitro, the combination led to greater tumour progression and animal survival [263]. THZ1 was also combined with some other inhibitors from different groups, for example with the HDAC inhibitor panobinostat in diffuse intrinsic pontine glioma. This combination leads to a synergetic reduction of cell viability. It was shown that THZ1 and panobinostat together disrupt SE biology [271]. In neuroblastoma, THZ1 and panobinostat also show synergy downregulating JMJD6, E2F2, N-Myc, and c-Myc expression in cell lines and in vivo [272]. There are some examples of combinations of SE inhibitors with chemotherapy. For example, promising in vivo and in vitro results have been obtained for cisplatin-resistant ovarian cancer using cisplatin in combination with a BET inhibitor [274].

Just as the study of super-enhancer regions can help in the discovery of new drugs, the use of already conventional anticancer drugs helps shed light on the nature of malignancy and the involvement of SEs in this process. For example, researchers studied the effects of cisplatin, an anticancer drug administered at suboptimal and intermittent doses to avoid life-threatening effects. Cisplatin is interesting because, although this regimen improves symptoms in the short term, it also leads to more malignant disease in the long term. Using experimental data obtained on ovarian carcinoma cells using a PageRank-based algorithm, the super-enhancer regulator ISL1 is predicted to be the driving force behind this plasticity. The prediction was experimentally confirmed using the tools of CRISPR–dCas9–KRAB inhibition (CRISPRi) and CRISPR–dCas9–VP64 activation (CRISPRa), confirming the hypothesis of cisplatin reprogramming of cancer cells and explaining its pro-oncogenic effect [150].

Histone modifications, including those that are markers of SEs, are introduced by epigenetic writers, such as histone deacetylases (HDAC) and histone demethylases (for example, lysine-specific histone demethylase 1) [275]. Histone acetylation is coregulated by the enzymes histone acetyltransferase and HDAC; both are in a state of dynamic equilibrium. Ample evidence suggests that the malignant phenotype of GBM is regulated by SE and HDAC. HDAC1, along with BRD4, Pol II, and other key components, is enriched in SE [17]. LSD1 is indirectly involved in the deacetylation of H3K27ac [276] and eliminates SE activity [277].

We have reviewed the main strategies for super-enhancer suppression using BETi, CDK7i, and HDACi. These groups of inhibitors are very extensive, and experimental studies of the effect on the functioning of SEs have been carried out not for all of their representatives, which is of interest and needs further study. Some of the inhibitors considered (for example, such BETi, as OTX015, BAY1238097, iBET-151, and volasertib), despite being effective in cell lines and xenograft models, have not been clinically tested due to toxicity, side effects, and other reasons. However, in recent years, new inhibitors from the above groups have been investigated, some of which are undergoing clinical trials. There are also super-enhancer inhibitors approved by the US FDA, for example, such HDACis as romidepsin, vorinostat, panobinostat and belinostat. HDACis have proven to be the safest group of SE inhibitors to date. Many researchers see the combination of various super-enhancer inhibitors as a promising strategy as well as their combinations with chemotherapy, which can potentially improve the effectiveness of therapy due to the complex effect.

## 5. Conclusions

This review aims to provide a comprehensive examination of the current state of SE research, delineating its achievements while shedding light on areas of ambiguity and ongoing academic debate. While there is a consensus regarding the significant role of SEs, the exact modality of their function remains a topic of discussion among researchers.

Divergence in findings pertains to whether SEs have an additive effect or engage in synergistic interactions. The potential role of noncoding super-enhancer RNAs in the regulatory processes accentuates the need for rigorous investigation in this domain. Our assessment also pointed to the challenges in SE classification and prediction. For example, the predictive value of H3K27ac is under question because some SEs are enriched in H3K122ac but not in H3K27ac. Using H3K27ac solely to predict SE can lead to underprediction of SE [7].

Despite numerous attempts at categorising SEs based on their functionalities, there remains a lack of uniformity in classifications. Furthermore, the overlap between locus-control regions (LCRs) and SEs has yet to yield clear, consistent patterns. The observed evolutionary conservation of SEs further necessitates a deeper dive into their evolutionary role and relevance.

From a clinical perspective, the involvement of super-enhancers in pathological conditions, particularly their formation in oncogenic processes, has crucial implications for therapeutic research. The role of SE in oncological diseases was also discussed in more detail in the following reviews [44,45,172,278,279,280,281,282]. SEs’ roles in inflammatory processes further extend their relevance to a wide range of diseases.

If a super-enhancer is discovered to be associated with the disease and related pathological processes, gene expressions modulated by this SE can be employed as disease biomarkers and medication efficacy indicators. Furthermore, the revealed connection is a motivation to study, for example, BET inhibitors as the first line of epigenetic therapy for this condition.

BET inhibitors are currently the most versatile approach to SE-targeted therapy since SE occupancy by BRD4 is always required for the activity of super-enhancers, which are more sensitive to this regulator compared to typical enhancers. In cases where BET inhibitors are ineffective, therapy with small-molecule inhibitors targeted at other proteins or a combination of proteins in the SE complex should be considered. SE inhibitors are being actively studied to combat oncological diseases, among which are BET, CDK7, and MYC inhibitors (for cancer types in which an abnormally high expression of the MYC gene is observed), as well as their combinations with chemotherapy (e.g., a combination therapy with cisplatin and JQ1 for cisplatin-resistant ovarian cancer has been suggested [183]).

However, as demonstrated above, many studies are at the phenomena level, and mechanism research, such as how SEs are regulated and the detailed molecular process by which SEs influence their target genes, is still lacking.

When known SE inhibitors do not work even in combination with other methods or show high toxicity and severe side effects in a given disease, genome and epigenome editing with CRISPR–Cas9 systems can be investigated. Although many reviewed papers suggest genome-wide changes in SE activity, it is possible to identify the SE with the strongest characteristic signal (e.g., ChIP-seq signal), often regulating the expression of master transcription factors and subsequently altering gene regulatory networks. Within a detected SE, it might be efficient to focus on specific SE constituent modules, which, when edited, will allow for gene-expression stabilisation to the needed levels. In addition, such editing can be a perspective approach in cases of SE loss.

The described approach for developing a therapeutic strategy is potentially applicable regardless of the mechanism of the SE changes in dynamics (formation, redistribution, gain, or loss). Nonetheless, knowledge of the exact mechanism may offer additional therapeutic options regarding key signalling molecules of the SE-correlated alterations in biological pathways.

Moreover, standard approaches to cancer subtyping rely heavily on expression-profile clustering, which cannot effectively differentiate oncogenic drivers and their secondary effects. At the current time, numerous studies based on SE profiles have found novel subgroups with unique biological characteristics and clinical implications. Additionally, established subtype-specific SEs and master regulators may offer novel biomarkers for cancer risk and therapeutic response prediction.

Hence, SEs might act as more illustrative molecular features of cancer subtypes because they are well-identified as critical elements defining cell identity. That may lead to a reorganisation of the existing cancer taxonomy.

Focusing on future research directions for SE investigation, there is a promising horizon where breakthroughs and evolving technologies could significantly enhance our grasp of SE complexities. Key areas of interest could include delving into the molecular bases of diseases, particularly cancer [283,284,285,286,287], probing molecular interactions with state-of-the-art optical methods [288,289,290,291], leveraging magnetic nanotags for the ultraprecise detection of biomolecules [292,293,294,295,296,297,298,299], applying advanced label-free biosensors [300,301,302,303,304], and utilizing cloud computing to analyse genome-wide data comprehensively [305,306]. These avenues not only promise to deepen our understanding of SE but also potentially redefine its conceptual framework.

In conclusion, as the field continues to advance, a clearer understanding of SE’s contribution to the pathogenic mechanisms will be crucial for both basic research and clinical applications.

## Figures and Tables

**Figure 1 ijms-25-03103-f001:**
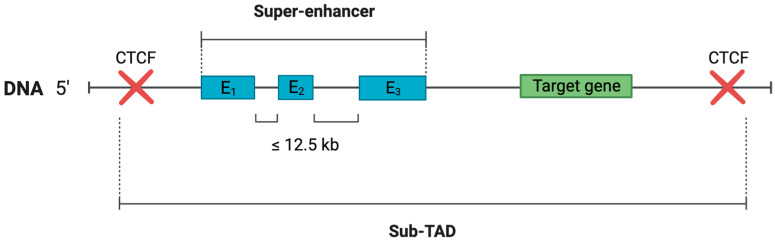
Sub-TAD structure scheme.

**Figure 2 ijms-25-03103-f002:**
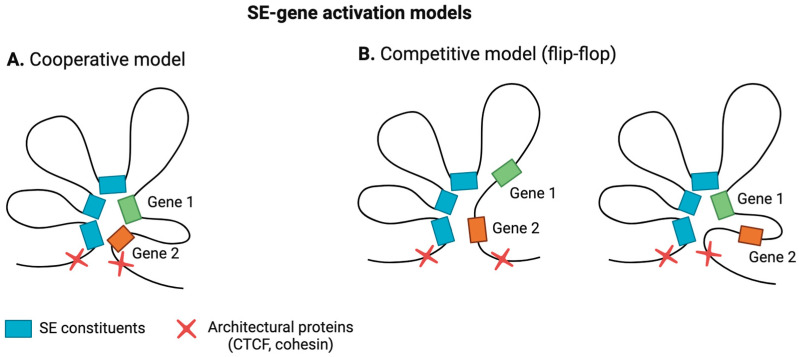
Models of SE-gene activation: (**A**)—cooperative; (**B**)—competitive (flip-flop).

**Figure 3 ijms-25-03103-f003:**
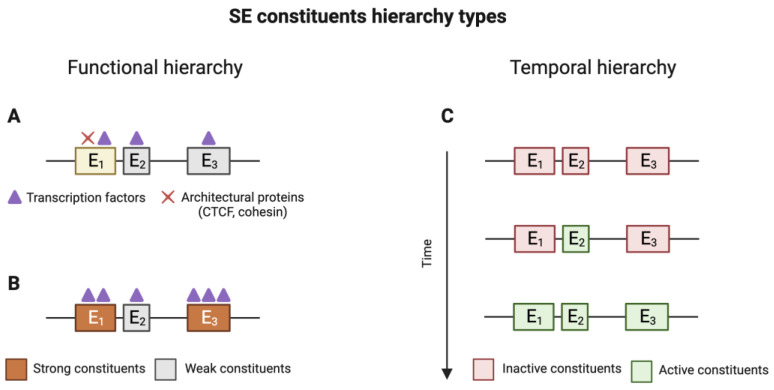
SE constituents’ hierarchy types. (**A**) Functional hierarchy—hub and non-hub enhancers. (**B**) Functional hierarchy—strong and weak substitutes. (**C**) Temporal hierarchy—active and inactive enhancers.

**Figure 4 ijms-25-03103-f004:**
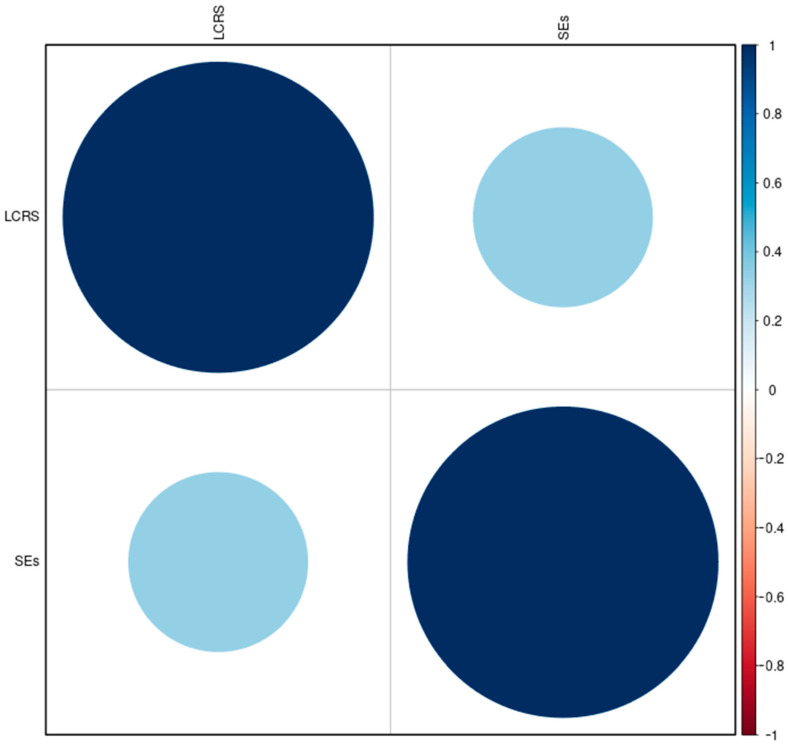
The presence of a consistent distribution pattern suggests an indeterminate functional association. The colours of circles in each cell represent co-occurrence correlation and can be numerically interpreted in accordance with a scale at the right Y-axis. According to the temperature scale (on the right) deep blue depicts high correlation and light blue represents low correlation between SE and LCR loci. The sizes of the circles represent the amount of analysed entities within every group.

**Figure 5 ijms-25-03103-f005:**
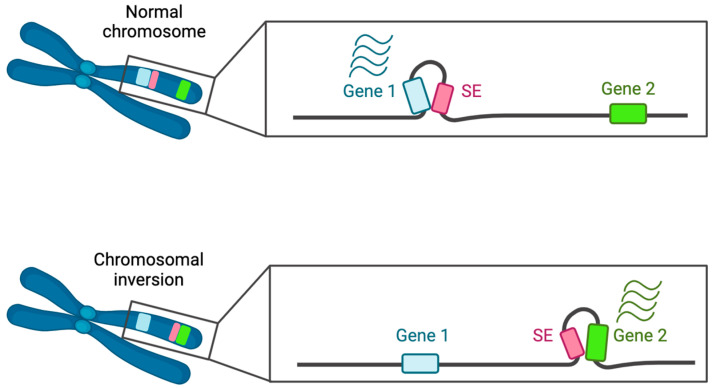
Example of chromosomal rearrangements: chromosomal inversion, leading to SE repositioning.

**Figure 6 ijms-25-03103-f006:**
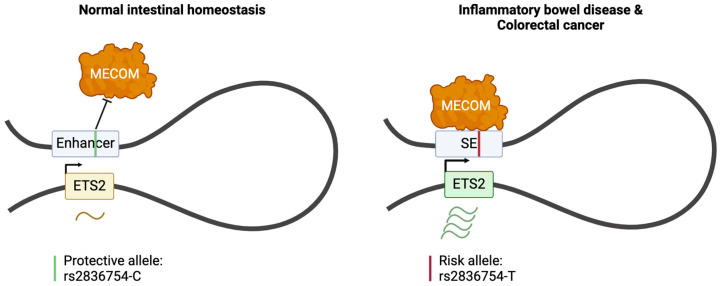
Example of SNP’s influence on SE dynamics. Green line—schematic representation of a protective allele (rs2836754-C) preventing MECOM from effective SE activation of ETS2 transcription; red line—schematic representation of a risk allele (rs2836754-T), accelerating *ETS2*-SE.

**Figure 7 ijms-25-03103-f007:**
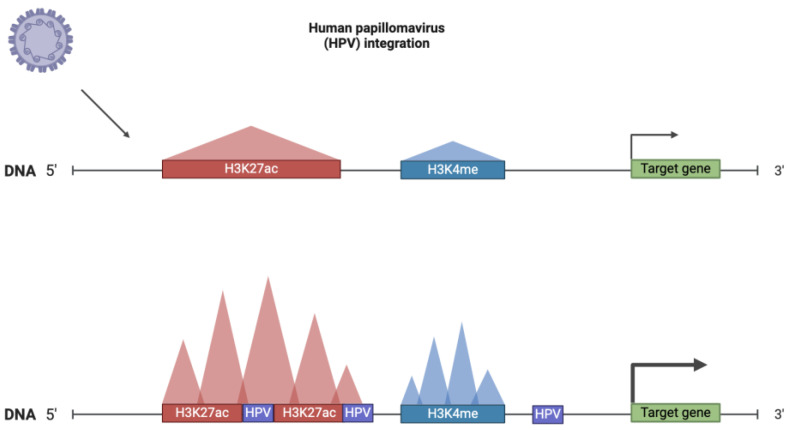
Example of viral integration (HPV) influence on SE dynamics.

**Table 1 ijms-25-03103-t001:** SEs and stretched enhancers differences.

Characteristics	Super-Enhancers	Stretched Enhancers
Number	x ^1^	10x
Genome coverage	x	2x
Average size	22,812 bp	5060 bp
Distance from TSS	<2 kb (69% of SE)	>10 kb (70% of stretched enhancers)
Evolutionary conservation (phastCons scores for 99 vertebrate genomes)	more conserved	less conserved
H3K27ac	enriched	depleted
H3K4me3	enriched	depleted
H3K4me1	enriched	enriched
H3K27me3	depleted	enriched
DHSs	higher	lower
Cohesin (SMC3 and RAD21 components)	more	less
CTCF	more	less
RNA II	more	less
Expression of associated genes	higher	lower
eRNA	more	less
Comparison between cell types	less shared	significantly shared

^1^ Depends on the cell line.

**Table 2 ijms-25-03103-t002:** SE databases comparison.

Database	dbSUPER	Sedb v.2.0	SEA v.3.0	Seanalysis 2.0
Link	https://asntech.org/dbsuper/ (accessed on 14 July 2023)	https://bio.liclab.net/sedb/ (accessed on 14 July 2023)	http://sea.edbc.org/ (accessed on 14 July 2023)	https://bio.liclab.net/SEanalysis/ (accessed on 14 July 2023)
Last update	2017	2022	2019	2023
Organisms	Human, mouse	Human, mouse	Human, mouse, *Drosophila melanogaster*, *Caenorhabditis elegans*, chicken, chimp, rhesus, sheep, *Xenopus tropicalis*, stickleback, zebrafish	Human, mouse
Data source (ChIP-Seq)	Collected existing data on SEs from published articles + NCBI GEO	NCBI GEO/SRA, ENCODE, Roadmap, Genomics of Gene Regulation Project (GGR), and National Genomics Data Center Genome Sequence Archive (NGDC GSA)	NCBI, GEO/SRA, ENCODE	Data from SEdb
Search pipeline (ChIP-Seq data alignment, peak calling, SE search, accordingly)	Varied between articles included in the database, more details in [95]	Bowtie2, MACS2, ROSE	Bowtie2, MACS2, ROSE	
Human				
Input data for SE prediction	H3K27ac	H3K27ac	H3K27ac, p300, BRD4, Med1	H3K27ac
Genome version	hg19	hg38, hg19	hg38	hg19
Mouse				
Input data for SE prediction	Med1 (for ESCs and pro-B cells), MyoD (myotubes), T-bet (Th cells), C/EBP (macrophages)	H3K24ac	H3K24ac, p300	H3K27ac
Genome version	mm9	mm39, mm10	mm10	—
Database completeness				
Human				
SEs	68,729	1,167,518	109,304	1,167,518
Tissue types	Not given	118	Not given	Not given
Cell lines	Not given	Not given	133	Not given
Tissue types/cell lines	99	Not given	141	∼180
Mouse				
SEs	2558	550,226	23,969	550,226
Tissue types	Not given	636	Not given	19
Cell lines	Not given	1107	Not given	—
Tissue types/cell lines	5	Not given	32	∼110
Annotation completeness				
Tissue type	Yes	Yes	Yes	Yes
SE genomic location	Yes	Yes	Yes	Yes
SE constituents	Yes	Yes	Yes	Yes
SE conservation	No	Yes	Yes	Yes
Associated genes	Yes	Yes	Yes	Yes
RNA-seq	No	Yes	No	
DNase I hypersensitive sites (DHSs)	No	Yes	No	Yes
Methylation sites	No	Yes	Yes	No
H3K27ac	No	Yes	Yes	Yes
Chromatin interactions region	No	Yes	Yes	No
TADs	No	Yes	Yes	No
TF binding sites (predicted)	No	Yes	Yes	Yes
TF binding sites (ChIP-Seq)	No	Yes	Yes	Yes
TF binding sites conservation	No	Yes	Yes	Yes
SE associated pathways	No	No	Yes	Yes
SNPs	No	Yes	Yes	Yes
eQTL	No	Yes	No	No
CRISPR–Cas9 target site	No	Yes	Yes	No
Visualisation				
Genome browser	UCSC	On the site	On the site	USCS, on the site
SE location	Yes	Yes	Yes	No
SE constituents’ location	No	Yes	Yes	No
Associated genes	No	Yes	Yes	Yes
Nearby genes	Yes	Yes	?	Yes
Genes expression	Yes	Yes	Yes	Yes
Enhancers	No	Yes	No	No
Methylation sites	Yes	Yes	Yes	Yes
H3K27ac	Yes	No	Yes	Yes
p300	No	No	Yes	No
BRD4	No	No	Yes	No
Med1	No	No	Yes	No
TF binding sites	No	Yes	Yes	No
Chromatin interactions region	Yes	Yes	Yes	Yes
Regulatory networks formed by SE (SE, TF, gene)	No	Yes	Yes	Yes
SE conservation	Yes	Yes	Yes	Yes
SNPs	Yes	Yes	Yes	Yes
CRISPR–Cas9 target site	Yes	No	Yes	Yes
Analytic tools	Overlap analysis, downstream analysis (Galaxy server), Gene Ontology analysis (GREAT server), correlation analysis, gene-expression analysis and motif discovery (Cistrome server)	Differential Overlapping SE analysis, SE-based TF–gene analysis, Gene-SE analysis, SNP-SE analysis, Overlap analysis, Region analysis	GREAT (predicts functions of cis-regulatory regions), Enrichr (gene set enrichment analysis), Specific analysis of H3K27ac status, SE cell-type specificity, TF enrichment analysis, Regulatory network	Pathway downstream analysis, Upstream regulator analysis, Genomic region annotation TF regulatory analysis, Sample Comparative analysis
Links with	Galaxy GREAT Cistrome	GREAT	Galaxy Enrichr	GREAT UCSC NCBI Gene GeneCards UniProt
Data submission option	Yes	No	No	No

**Table 4 ijms-25-03103-t004:** HDAC inhibitors with varying selectivity for HDAC isoforms [259,260,261,262].

Inhibitor Group	Inhibitor	Primary Targets	Secondary Targets
Benzamides	Entinostat (MS-275)	HDAC1/2	HDAC3
	Tacedinaline (C1994)	HDAC1/2/3	–
	Merck60	HDAC1/2	–
	Mocetinostat (MGCD0103)	HDAC1/2	HDAC3/11
	4SC202	HDAC1/2/3	–
Hydroxamic acids	Panobinostat	HDAC1/2/3/4/7/9	HDAC6/8
	Vorinostat (SAHA)	HDAC1/2/3/6/10/11	HDAC8
	Dacinostat (LAQ824)	HDAC1/2/3/6/10/11	HDAC4/5/8
	Rocilinostat (ACY1215)	HDAC6	Other HDACs
	WT161	HDAC6	Other HDACs
	Pracinostat (SB939)	All HDACs except 6	–
	MC 1568	Class IIa, HDAC6	Other HDACs
	OJI-1	HDAC8	–
Other	TMP195	Class IIa	–
	Selisistat (EX 527)	SIRT1	–
	Largazole	Class I, class IIb, class IV	–
	Romidepsin	Class I	–

**Table 5 ijms-25-03103-t005:** Combinations of inhibitors for cancer treatment.

Inhibitor 1	Inhibitor 2	Tumour	Effect	Reference(s)
JQ1	THZ1	Neuroblastoma	Synergetic inhibition of viability of neuroblastoma cell lines by CDK7 inhibition and BET inhibition	[263]
JQ1	THZ1	Pancreatic ductal adenocarcinoma	G2/M phase cell-cycle arrest, inhibition of cell migration and invasion by CDK7 inhibition and BET inhibition	[264]
JQ1	THZ1	Head neck squamous cell carcinoma	Antiproliferative and proapoptotic effects by CDK7 inhibition and BET inhibition	[265]
JQ1	YKL-5-124	Neuroblastoma	Synergistic cytotoxicity, CDK7 inhibition, and BET inhibition	[241]
JQ1	Milciclib	Medulloblastoma	Suppressing MYC-driven tumour, CDK2 inhibition, and BET inhibition	[266]
JQ1	ICG-001	Glioma	Strong cytotoxic effects, CBP and BET inhibition	[267]
JQ1	Trametinib	Colorectal cancer	Inhibition of cell proliferation by dual targeting of BET proteins and MAPK signaling	[244]
OTX015	CYH33	B-cell lymphoma	Cell-cycle arrest and apoptosis, BET inhibition combined with PI3Kα-selective inhibition	[268]
THZ1	Ponatinib/ lapatinib	Neuroblastoma	Cell apoptosis: CDK7 inhibition and tyrosine kinase inhibition	[269]
THZ1	BH3-mimetics	Glioblastoma	Growth reduction of tumours by CDK7 inhibition, and antiapoptotic effect	[270]
THZ1	Panobinostat	Diffuse intrinsic pontine glioma	Synergetic SE disruption, CDK7 inhibition, and HDAC inhibition	[271]
THZ1	Panobinostat	Neuroblastoma	Apoptosis induction in cancer cells, CDK7 inhibition and HDAC inhibition	[272]
HDAC inhibitors	FAO inhibitors	Glioblastoma	HDAC inhibition and influence lipid metabolism	[273]

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
