# Peer review of "Super-Enhancers and Their Parts: From Prediction Efforts to Pathognomonic Status"

_ijms, 2024, doi:10.3390/ijms25063103_

Round 1
Reviewer 1 Report
Comments and Suggestions for Authors
The manuscript provides a review of current advances in super enhancers, focusing on their classification and roles in cancers and other diseases. The contents are detailed, and the information is comprehensive. However, there are excessive details in some areas, and certain content needs more clarity. Therefore, I recommend some major revisions before publication. Here are specific points for consideration:
1. There are some missing references, e.g., the first and second paragraphs in the Introduction, lines 75-82, 171-180, 191-192, 238-240, 250-253, etc.
2. Clearly explain abbreviations, such as "BET" and "LCR" in Section 2.1, "CTCF" in Section 2.2, "DN" in Section 2.7.1, etc., when first introduced to enhance readability.
3. Avoid introducing information unrelated to the specified subtitles. Review and ensure that content, such as lines 86-104, 191-208, 817-864, etc., aligns with the provided context.
4. There is a lack of connection between contents. For example,
a) In Section 2.2 (lines 146-155), the transition from "One critical aspect of SEs function lies in their interaction with target gene promoters, a process instrumental in shaping gene expression patterns crucial for normal cellular function and often disrupted in disease states." to "Firstly, the strength of topologically associated domains (TAD) boundaries is believed to ensure the target operation of SEs." is abrupt and lacks coherence.
b) Section 3.3 needs an overview paragraph to lead the entire section.
c) Table 3 should be mentioned earlier in Section 4.
5. For Table 2,
a) Clearly explain the contents, for example, there are two identical "input data for SE prediction" and "Genome version," and the reason why they have different contents following needs to be explained.
b) It can be moved to the Appendix to enhance readability.
6. The figure captions in Figure 4 should explain the figures clearly, including the meaning of the area of the circles, and the meaning of color temperatures.
7. For Section 3.1, the authors listed detailed examples about super enhancers in oncology, but too much detail and too boring. Please consider reorganizing this section.
8. For Table A1,
a) Consider to reorganize or split the table to enhance readability. It is too large to find information effectively.
b) Check the upper right blank in page 40.
Comments on the Quality of English LanguageSome sentences, such as those in lines 159-161, 724-726, 873-875, and 1007, are challenging to understand.
Reviewer 2 Report
Comments and Suggestions for Authors
The publication is devoted to the theme of the superenhancers. The authors analyzed the history of the study of these regulatory elements, compared them with other enhancer sequences, analyzed the classification of superenhancers, their structure and mechanisms of action. The role of superenhancers in the pathogenesis of various diseases is analyzed, the authors describe some therapies targeting superenhancers. A large summarizing table on the role of superenhancers in cancer is given.
The authors have performed a lot of work and analyzed a considerable amount of literature. The publication analyzes and compares the characteristics of superenhancers and stretched enhancers. This is an important contribution as currently no unified classification of these elements has been developed. The authors also provide a good overview of available databases on superenhancers.
The major criticism of this publication is that it is very poorly illustrated. Publications of this type should be accompanied by more illustrative material as well as graphical abstracts.
The second observation is the language of the publication. Authors need to be more careful in formulating their thoughts in English. For example, it is better to replace anti-SE therapy with SE-targeted therapy. On line 133 "formation of SEs" - does it mean formation during evolution or functional activation of superenhancer?
Reviewer 3 Report
Comments and Suggestions for Authors
The article titled "Super-enhancers and their parts: from prediction efforts to pathognomonic status" by Anastasia V. Vasileva et al. offers valuable insights. However, there are several issues that need to be addressed:
1. Author must write the aim of the study in the abstract
2. The introduction provides a general overview of super-enhancers and their role in gene regulation, but lacks a clear statement of the research gap or the specific objectives of the review.
3. The statement about predicting super-enhancer dynamics based on enhancer hierarchy and aggregated action is intriguing, but it is not clear how this prediction can be achieved or what evidence supports this idea.
4. The mention of LCRs is interesting, but the connection and differentiation between super-enhancers and LCRs is not fully explained.
5. While the review promises to summarize the classification and interaction of super-enhancer constituents, it is not clear what specific constituents are being referred to or how their interaction and cross-regulation are explored.
6. The review mentions the involvement of super-enhancers in oncological and immunological diseases, but it is not clear how the specific mechanisms and diversity of super-enhancers in these diseases will be addressed.
7. The conclusion of the review states that it will provide a detailed overview of super-enhancers in different diseases and act as a rationale for future clinical interventions. However, it would be helpful to briefly mention the potential implications of understanding super-enhancers in disease pathogenesis and how this knowledge could lead to targeted therapeutic interventions.
Round 2
Reviewer 1 Report
Comments and Suggestions for Authors
The revised vision has significantly improved and flows well. I really appreciate the addition of figures in Section 3.1. I believe it is now suitable for publication.
Reviewer 2 Report
Comments and Suggestions for Authors
The authors have taken into account the comments made. Figures have been added, and some phrases in the text have been corrected.
The manuscript can be recommended for publication.
Reviewer 3 Report
Comments and Suggestions for Authors
Accept in present form
Comments on the Quality of English LanguageModerate editing of English language required